# H3K4 methylation at active genes mitigates transcription-replication conflicts during replication stress

Shin Yen Chong [ID] [1,2], Sam Cutler[3], Jing-Jer Lin [ID] [4], Cheng-Hung Tsai [ID] [5], Huai-Kuang Tsai [ID] [5], Sue Biggins [ID] [3,6], Toshio Tsukiyama[3], Yi-Chen Lo [ID] [2]* & Cheng-Fu Kao [ID] [1]*

Transcription-replication conflicts (TRCs) occur when intensive transcriptional activity compromises replication fork stability, potentially leading to gene mutations. Transcription-deposited H3K4 methylation (H3K4me) is associated with regions that are susceptible to TRCs; however, the interplay between H3K4me and TRCs is unknown. Here we show that H3K4me aggravates TRC-induced replication failure in checkpoint-defective cells, and the presence of methylated H3K4 slows down ongoing replication. Both S-phase checkpoint activity and H3K4me are crucial for faithful DNA synthesis under replication stress, especially in highly transcribed regions where the presence of H3K4me is highest and TRCs most often occur. H3K4me mitigates TRCs by decelerating ongoing replication, analogous to how speed bumps slow down cars. These findings establish the concept that H3K4me defines the transcriptional status of a genomic region and defends the genome from TRC-mediated replication stress and instability.

---

[1] Institute of Cellular and Organismic Biology, Academia Sinica, Nankang, Taipei 11529, Taiwan. [2] Graduate Institute of Food Science and Technology, National Taiwan University, Taipei 10617, Taiwan. [3] Basic Sciences Division, Fred Hutchinson Cancer Research Center, Seattle, WA 98109, USA. [4] Institute of Biochemistry and Molecular Biology, National Taiwan University College of Medicine, Taipei 10051, Taiwan. [5] Institute of Information Science, Academia Sinica, Nankang, Taipei 11529, Taiwan. [6] Howard Hughes Medical Institute, Division of Basic Sciences, Fred Hutchinson Cancer Research Center, Seattle, WA 98109, USA. *email: loyichen@ntu.edu.tw; ckao@gate.sinica.edu.tw

Transcription-replication conflicts (TRCs) can significantly contribute to genome instability[1–4]. These naturally mutagenic impediments to ongoing replication result from transcription and replication machineries simultaneously operating on the same section of a DNA template[5,6]. By stalling the replication fork, TRCs predispose DNA to mutations, recombination, and strand breaks[5,7,8]. In prokaryotes, TRCs are likely prevalent, functioning as an important evolutionary source of mutagenesis in stress-response genes[9–13]. However, eukaryotic transcription and replication machineries encounter each other only under specific circumstances. For example, in *Saccharomyces cerevisiae*, DNA-polymerases are sometimes stalled at highly transcribed RNA pol II genes[8] and RNA pol III-transcribed tRNA genes[14]. In human cells, transcription of the longest genes may last longer than a single cell cycle, and these long genes only commence replication in late S phase. Some such genes exist in common fragile sites (CFSs), which are hotspots for chromosomal fragility due to relatively frequent encounters between replication forks and transcription elongation complexes[7]. Early-replicating fragile sites (ERFSs) are another class of fragile sites found in actively transcribed and early replicated regions of mouse and human DNA[15], and conflicts between transcription and replication machinery may also account for the fragility of ERFSs.

The ataxia telangiectasia and Rad3-related protein (ATR)-dependent S-phase checkpoint promotes completion of DNA synthesis by regulating origin firing, stabilizing replication forks, and promoting fork repair and restart in mammalian cells[16]. ATR kinase regulates the stability of both CFSs and ERFSs, suggesting that the S-phase checkpoint is important for monitoring ongoing replication and resolving TRCs[2]. In yeast, the Mec1$^{ATR}$ and Rad53 signaling pathways are crucial for resolving TRCs. For example, to reduce the catastrophic effects of TRCs, Rad53 is activated by Mec1 to uncouple transcribed genes from nuclear pores, thereby releasing the topological tension caused by dual activity of replication and transcription complexes[17].

Members of the MLL (mixed-lineage leukemia) histone-lysine *N*-methyltransferase (also known as KMT2) family methylate histone H3 on lysine 4 (H3K4) in mammals[18]. Translocations of MLL1 can result in the creation of oncogenic fusion proteins (i.e., MLL1, ALL-1, and HRX) that drive a subset of infantile and adult leukemias[19,20]. Methylated H3K4 (H3K4me) is widely recognized as an activating histone modification[21,22], and the enzymes that perform this methylation are highly conserved[18]. In *Saccharomyces cerevisiae*, the Set1 complex (Set1C/COMPASS) site-specifically methylates H3K4—forming mono-methylated, di-methylated, or tri-methylated residues (H3K4me1/2/3)—by the action of seven regulatory components that physically associate with the Set1 protein[23] (Supplementary Fig. 1a). In this process, H3K4me3 is enriched at transcription start sites, followed by downstream deposits of H3K4me2 and H3K4me1, creating a 5′ to 3′ gradient of H3K4 methylation. Despite the fact that H3K4me3 is considered to be an activation marker[22,24], its functional role is not firmly established. Loss of H3K4me3 has only minor effects on transcription for the majority of genes in yeast and mammals[25,26], and some evidence has even suggested that H3K4me on genes may be the result, not the cause, of transcription, with the gradient of H3K4me on genes arising from multiple rounds of transcription[27]. Therefore, the exact role of H3K4me in regulating transcription remains to be defined[26,28].

A hint that H3K4 methylation may play a role in replication comes from the observations that H3K4me is actively erased during replication[29] and is associated with the response to replication stress[30]. Deletions of Set1C subunits or H3K4 mutations increase sensitivity of yeast cells to high-dose HU[31], and SETD1A-deposited H3K4me stabilize stalled forks, preventing them from degradation in HeLa cells[32]. Together these observations imply that H3K4 methylation may be crucial to protect cells from replication perturbations. In the current study, we report that *RAD53* is epistatic to *SET1*-dependent H3K4me, which contributes to the full function of the S-phase checkpoint by preventing TRCs. Using a system for inducing TRCs, we found that ablation of H3K4me allows replication forks to progress in the face of strong transcription with a cost of predisposing the genome to mutations. Our results thus demonstrate a previously unknown role of H3K4me in TRC prevention and provide further mechanistic insight into the role of chromatin during DNA replication stress.

## Results

**H3K4me ablation rescues the *rad53* viability under HU stress.** Budding yeast cells lacking Rad53 checkpoint activity are hypersensitive to low doses of HU, likely because of stalled replication forks[17,33,34]. We previously discovered that cell viability of HU-treated *rad53* mutants was improved by abolishing mono-ubiquitylation of H2B (H2Bub)[35]. To understand the underlying mechanism of this protection, we tested whether the HU sensitivity of *rad53* mutants was determined by events downstream of H2Bub, specifically Set1-mediated H3K4 and/or Dot1-mediated H3K79 methylation[36,37]. To this end, we introduced several H3 mutations into *rad53* cells and discovered that H3K4A (lysine at position 4 of H3 was substituted with alanine) specifically increased the viability of *rad53* mutants during chronic HU stress (Fig. 1a). Notably, the H3R2A mutation also rescued HU sensitivity of *rad53* mutants, but to a lesser extend than H3K4A. It was previously reported that the H3R2A mutation reduces H3K4me3, but H3K4me2 and H3K4me1 remain unaltered[38], suggesting that the extent of H3K4 methylation influences HU sensitivity of *rad53* mutants. Because Set1 forms a complex (Set1C/COMPASS) that mediates methylation of H3K4[23] (Supplementary Fig. 1a), we were able to further manipulate H3K4 methylation levels by another method. As such, we introduced *set1Δ* (lost H3K4me1/2/3), *bre2Δ* (loss of H3K4me3 and reduced H3K4me2) or *spp1Δ* (reduced H3K4me3) into *rad53* mutants (Fig. 1b)[39]. Intriguingly, the efficiency of HU-sensitivity rescue in *rad53* mutants was negatively correlated with the level of H3K4 methylation in *set1Δ*, *bre2Δ*, and *spp1Δ* upon chronic or acute HU treatments (Fig. 1c, d). Moreover, the HU-sensitivity rescue effect by H3K4me ablation was generally represented in Rad53 checkpoint kinase-dead cells, as demonstrated with a different *RAD53* mutated allele or null mutation (Supplementary Fig. 1c, d).

Next, to test the possibility that loss of H3K4me may relieve fork stalling in *rad53* cells[34], we assessed the status of phosphorylated H2A-S129 (γH2A), which normally accumulates in genomic loci that are prone to replication-fork stall[40]. We found that the global level of γH2A in HU-treated *rad53* mutants was significantly decreased by the deletion of Set1C subunits (Fig. 1e), suggesting that the presence of H3K4me in the *rad53* background correlates with increased replication fork stalling. Interestingly, we found that the H3K4A mutation showed no effect on cell viability of *mec1Δ* cells in HU, indicating that the interaction between H3K4me and *RAD53* is uncoupled from Mec1/ATR signaling (Supplementary Fig. 1b).

One possible mechanism by which H3K4me influences *rad53*-HU sensitivity is through a transcription-linked process, such as R-loop formation or the accumulation of aberrant replication fork intermediates. We found that deleting genes encoding the RNase H components, *RNH1* and *RNH201*, sensitized *rad53* mutants to HU stress; however, the loss of H3K4me was able to rescue HU-treated *rad53* cells even in the absence of *RNH1* and *RNH201*, a condition that is known to result in the accumulation of

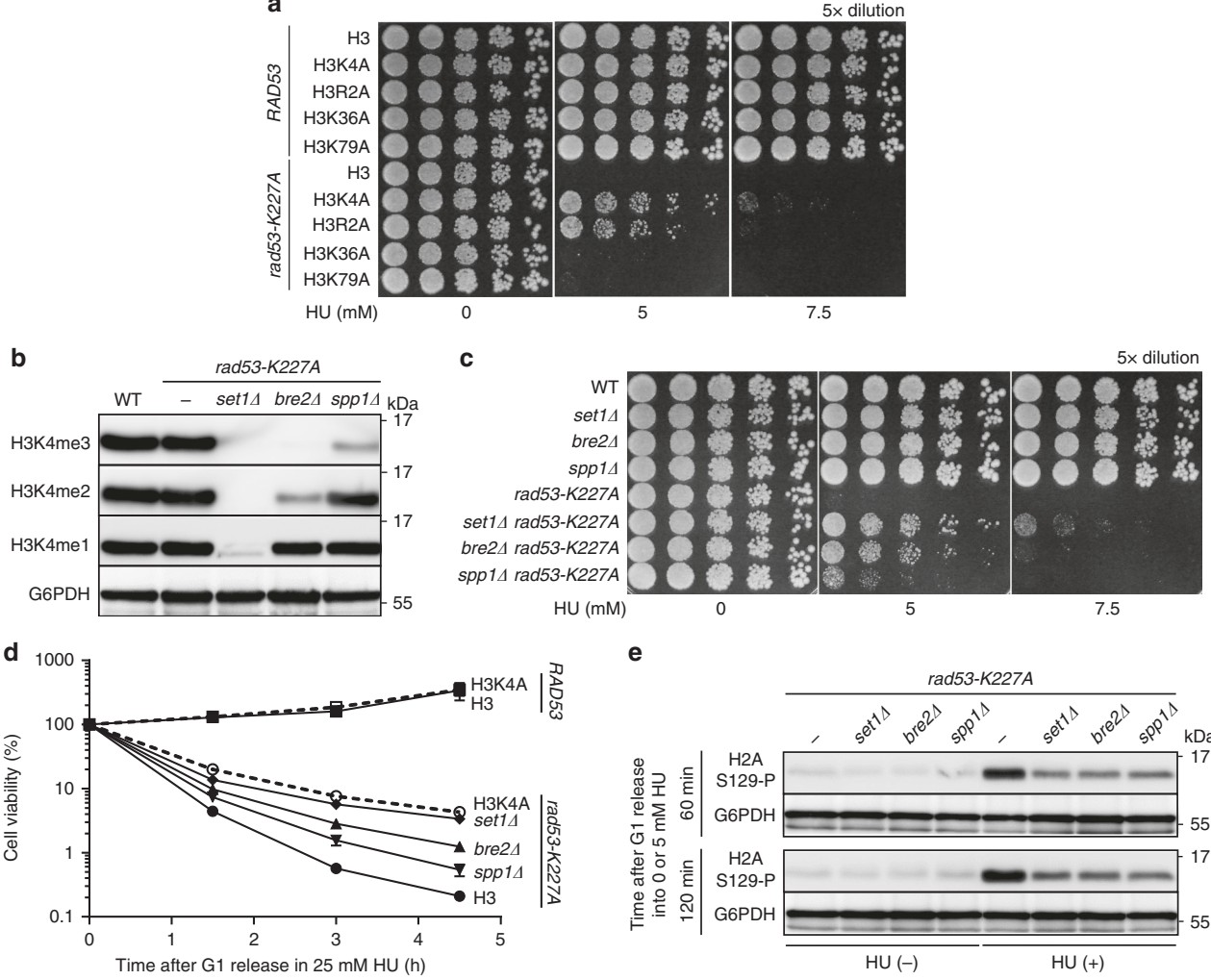

**Fig. 1 Decreased H3K4 methylation level improves viability of *rad53* mutants exposed to HU-induced replication stress. a** HU sensitivity was determined with H3 (WT), H3K4A, H3R2A, H3K36A and H3K79A mutations in the *RAD53* or *rad53-K227A* backgrounds. **b** Immunoblot showing H3K4me1/2/3 levels in *set1Δ bre2Δ* and *spp1Δ* strains. **c** HU sensitivity was determined with WT, *set1Δ, bre2Δ,* and *spp1Δ* in the *RAD53* or *rad53-K227A* backgrounds. **d** Cell viability after G1 synchronization and release into 25 mM of HU for the indicated time for H3, H3K4A, *set1Δ, bre2Δ* and *spp1Δ* on *RAD53* or *rad53-K227A* backgrounds. Data are the mean ± SD (*n* = 3 biological replicates). **e** Immunoblot showing levels of phosphorylated H2AS129 (γH2A) in *set1Δ, bre2Δ* and *spp1Δ* on the *rad53-K227A* background after G1-release in 0 mM (−) or 5 mM HU (+) for the indicated time.

R-loops[41,42] (Supplementary Fig. 2a). This observation suggests that the effect of H3K4 methylation is independent of R-loops, which are formed during transcription and have been linked to genome instability[10,43,44]. In addition, we found that genes involved in coupling transcribed genes to the nuclear pore[17] and the Rrm3 DNA helicase, which promotes fork reversal in *rad53* defective cells[34], also did not contribute to the interaction between H3K4me and *RAD53* (Supplementary Fig. 2b, c). Taken together, these genetic analyses indicated the ablation of Set1-dependent H3K4me is epistatic to the *RAD53* mutation and the level of their genetic interaction is negatively correlated with the level of H3K4 methylation.

**H3K4me ablation restores DNA Pol2 binding in *rad53* mutant.**
To investigate how H3K4 methylation influences the processes of transcription and replication under HU-stress, we used chromatin immunoprecipitation-sequencing (ChIP-seq) to identify genomic sites of active transcription marked by Rpb3, a subunit of RNA Polymerase II, and sites of replication pausing marked by DNA Pol2, a subunit of DNA Polymerase ε[8]. Neither Rad53

kinase function nor H3K4 methylation had a significant effect on global Rpb3 binding profiles in open reading frames (ORFs) (Fig. 2a) after exposure to 25 mM HU for 2 h. We further analyzed the Rpb3 distribution by dividing all ORFs into quintiles according to their expression efficiency (see Methods section). The Rpb3 binding patterns were highly similar in all quintiles among the four strains (Fig. 2b). Similar Rpb3 occupancies were also observed at several selected genes that are highly transcribed (Fig. 2c). The analysis of Spearman's rank correlation coefficient reinforced the notion that the loss of H3K4 methylation only had very minor effects on genome-wide Rpb3 distribution; the coefficient values were between 0.95 and 0.96 for pairwise comparisons between H3 and H3K4A mutant cells (Fig. 2d, left panel). These results support the argument that H3K4 methylation does not play a major role in regulating transcription[25,26,28].

In contrast to Rpb3 binding, DNA Pol2 binding was significantly affected by both *RAD53* and H3K4me. DNA Pol2 was paused at highly transcribed ORFs in WT (*RAD53*-H3) cells, as found in previous studies[5,8,45]. Interestingly, the enrichment of DNA Pol2 at these sites was slightly higher in H3K4A cells than in WT cells (Fig. 2a, b). This trend of increased DNA Pol2 levels

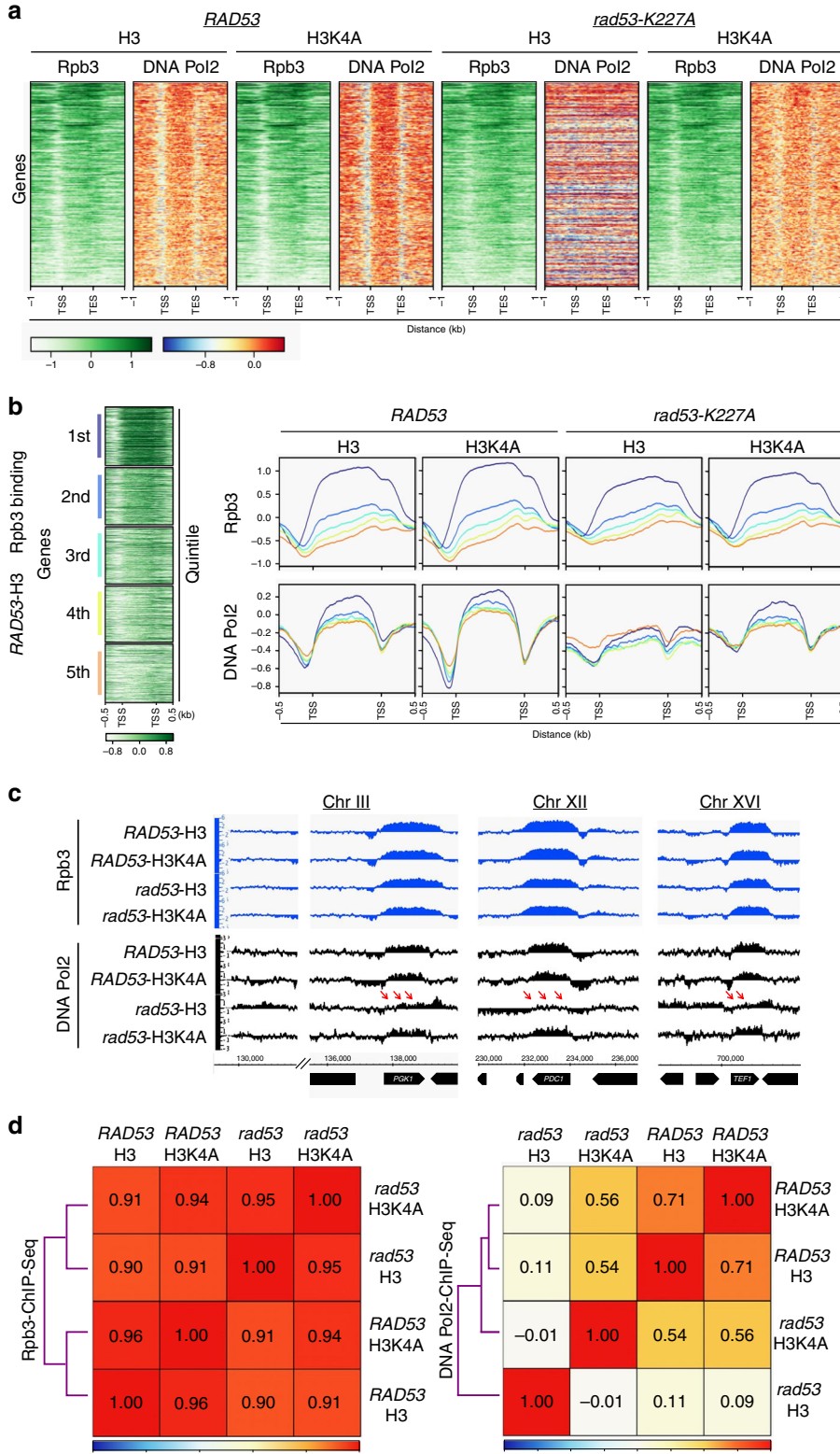

**Fig. 2 Loss of H3K4 methylation restores DNA Pol2 binding in *rad53* mutant under replication stress. a** Heatmaps of ChIP-seq signals for Rpb3 and DNA Pol2 from *RAD53*-H3, *RAD53*-H3K4A, *rad53-K227A*-H3 and *rad53-K227A*-H3K4A cells after 25 mM HU treatment for 2 h. TSS: transcription start site; TES: transcription end site. **b** Anchor plots of Rpb3 and DNA Pol2 ChIP signals sorted into quintiles, according to the expression efficiency (see Methods). Profiles in indicated colors represent gene clusters from the first to fifth quintile. Average signal strength of each cluster represented in log$_2$ scale. **c** Representative ChIP-seq tracks of Rpb3 and DNA Pol2 on selected highly transcribed genes in WT and mutant cells. *rad53* refers to the *rad53-K227A* mutant. The signals represent normalized sequence read density (log$_2$ scale) on loci along the indicated regions of respective chromosomes (see Methods). Red arrows mark the DNA Pol2 signals that were lower in *rad53*-H3 cells, indicating a lack of binding with regions undergoing transcriptional progression. **d** Spearman's rank correlation coefficient of genome-wide Rpb3 (right-panel) and DNA Pol2 (left-panel) ChIP-seq signals in indicated strains.

over ORFs in H3K4A cells was observed in all quintiles of transcriptional efficiency (Fig. 2b). However, the binding of DNA Pol2 was totally lost in HU-treated *rad53*-mutant cells (Fig. 2a, b). The loss of DNA Pol2 occupancy was more evident at highly transcribed ORFs, while Rpb3 binding at these sites was not affected (Fig. 2c). Surprisingly, the DNA Pol2 occupancy was restored in *rad53* mutants by the loss of H3K4 methylation (Fig. 2a). This restoration was particularly noticeable at highly transcribed ORFs (Fig. 2b, c). The pairwise Spearman correlation of DNA Pol2 genome-wide binding patterns was calculated (Fig. 2d, right panel), yielding a coefficient of 0.11 between WT and *rad53*-mutant cells. Such a low coefficient indicated exceedingly dissimilar DNA Pol2 binding patterns. In contrast, the coefficient between WT and H3K4A single mutants was 0.71, and the coefficient between WT and *rad53*-H3K4A double mutants was an intermediate value of 0.54. This analysis supported the idea that loss of H3K4me largely restored DNA Pol2 binding genomic regions with elevated transcriptional activity in *rad53*-mutant cells.

**H3K4me causes fork stalling in *rad53* mutants under HU stress**. There are two possible explanations for the observations in our genome-wide analyses. H3K4 methylation may be detrimental to *rad53* mutants by diminishing replisome stability at highly transcribed ORFs. Alternatively, this histone mark may impede the progression of replication forks beyond the regions immediately surrounding origins and into gene bodies. To distinguish these two possibilities, we investigated how H3K4 methylation influences DNA replication in *rad53* cells following replication stress by analyzing bulk DNA synthesis by fluorescence-activated cell sorting (FACS) analysis. We found that the replication stalling caused by HU in *rad53* cells was relieved by the H3K4A mutation or the deletion of *SET1*, *BRE2* or *SPP1* (Fig. 3a, purple slanted arrows), indicating that all three levels of H3K4 methylation influence fork progression. We then explored the effect of H3K4 methylation on replication progression over a chromatin template in 25 mM HU for 2 h. Mild accumulation of DNA Pol2 at late origins bound by the origin recognition complex (ORC) with little retention at early origins was observed (Fig. 3b) in WT (*RAD53*-H3) cells. We noted that the binding of DNA Pol2 showed a bimodal distribution in H3K4A cells with a deep trough centered at the autonomously replicating sequence (*ARS*), indicating that forks were moving away from the origins faster than those in WT. The progression of forks in *rad53* cells was clearly impaired, as DNA Pol2 largely accumulated at both early and late ORC-bound replication origins. This observation suggested that low concentrations of HU caused very slow-moving forks in the *rad53* mutant (Fig. 3b), consistent with the notion that in *rad53* mutants treated with HU, forks do not collapse but their progress is impaired[34]. In striking contrast, loss of H3K4 methylation in *rad53*-mutant cells nearly completely restored WT replisome dynamics in HU-treated (Fig. 3b and Supplementary Fig. 3a). These results suggest that H3K4 methylation becomes an impediment to the progressing replication forks in *rad53*-mutant cells under replication stress.

One possible mechanism by which decreased H3K4me alleviates fork stalling in HU-treated *rad53* mutants is that the modification may alter chromatin structure or dynamics. To test this possibility, we performed transposase-accessible chromatin sequencing (ATAC-seq)[46] to survey the genome-wide chromatin accessibilities in WT and mutant cells. We analyzed ATAC-seq data to profile open chromatin structures in HU-treated cells[47], finding that open chromatin regions in *RAD53* and *rad53* cells with either H3 or H3K4A mutation shared similar transposase accessibility profiles at transcription start sites (TSSs) (Fig. 3c).

Statistical analysis confirmed that the loss of H3K4 methylation only had very minor effects on chromatin structure at TSSs; the coefficient values for pairwise comparisons between wild-type H3 and H3K4A were 0.98 in *RAD53* and 0.97 in *rad53* backgrounds (Fig. 3d). Next, we examined the nucleosome fingerprint of the ATAC-seq pattern proximal to TSSs[46]. We observed a clear protection pattern from transposase insertion at sites immediately downstream of TSSs (Fig. 3e, purple arrows) in all strains, consistent with the known location of the +1 nucleosomes whose center is located 50–60 bp downstream of the TSS in yeast genes[48]. Following the +1 nucleosome-protected region, we observed similar periodic patterns of transposase protection in WT and mutant cells that may represent +2 and +3 nucleosomes (Fig. 3e and Supplementary Fig. 3b). Interestingly, we found that chromatin around ORC-bound replication origins became slightly less accessible by transposase in *rad53* cells compared to that in *RAD53* cells. However, the H3K4A mutation did not change the chromatin accessibility in either WT or *rad53* cells (Fig. 3f). Taken together, our ATAC-seq results showed that H3K4 methylation has no significant effect on global chromatin accessibility, suggesting that the restoration of replication progression seen in HU-treated *rad53* mutants without H3K4me is unconnected to alterations in chromatin structure.

**H3K4me ablation prevents defective forks in *rad53* mutants**. Because loss of H3K4 methylation prevents fork stalling in HU-treated *rad53* mutants (Fig. 3b), we reasoned that H3K4 methylation may contribute to HU-induced aberrant fork transitions in *rad53* mutants. To test this idea, replication intermediates emanating from *ARS305* were visualized by 2D gel after in vivo psoralen crosslinking of DNA digested by two combinations of restriction enzymes (Fig. 4a, b). H3 and H3K4A cells exhibited comparable 2D-gel profiles when treated with 25 mM HU (Fig. 4c, left panel). As expected, *rad53* cells accumulated X-shaped DNA recombinant intermediates, which correspond to fork reversal (Fig. 4c, purple triangles)[49]. Intriguingly, the signal representing reversed forks was significantly reduced in *rad53* H3K4A cells (Fig. 4c). Hence, these results indicated that H3K4me may contribute to fork reversal in *rad53* mutants treated with HU.

To further study how transcription-linked H3K4 methylation affects replication fork movement in *rad53* mutants under HU stress, we designed a system to induce head-on transcription-replication collisions. We inserted a methionine-inhibited promoter (pMET25) to drive *LRE1*, a gene located next to an early origin *ARS305* (Fig. 4d). In media lacking methionine, the *MET25* promoter yielded high expression levels of *LRE1*, along with enriched H3K4me3 deposition, which were comparable in all strains and had no effect on cell viability or replication progression upon HU stress (Supplementary Fig. 4a–d). WT and mutant cells were released from G1 in 25 mM HU and the DNA was analyzed by 2D gel. The activation of pMET25-LRE1 had no effect on fork progression in WT or H3K4A cells (Fig. 4e), but it strongly promoted fork reversal in *rad53* cells (Fig. 4f, upper panel: Met (−)). This augmentation of fork defects by active transcription was dramatically reduced in *rad53* H3K4A cells, though some residual aberrant fork structures were still observed (Fig. 4f, upper panel: Met (−)). Under repression of *pMET25-LRE1* (Fig. 4f, lower panel: Met (+)), *rad53* cells still accumulated X-shaped DNA recombinant intermediates, but these intermediates were diminished in *rad53* H3K4A cells, consistent with the idea that H3K4 methylation promotes fork abnormalities in *rad53* cells. Thus, our results support a model in which H3K4 methylation enhances the aberrant replication events caused by transcription-replication collisions

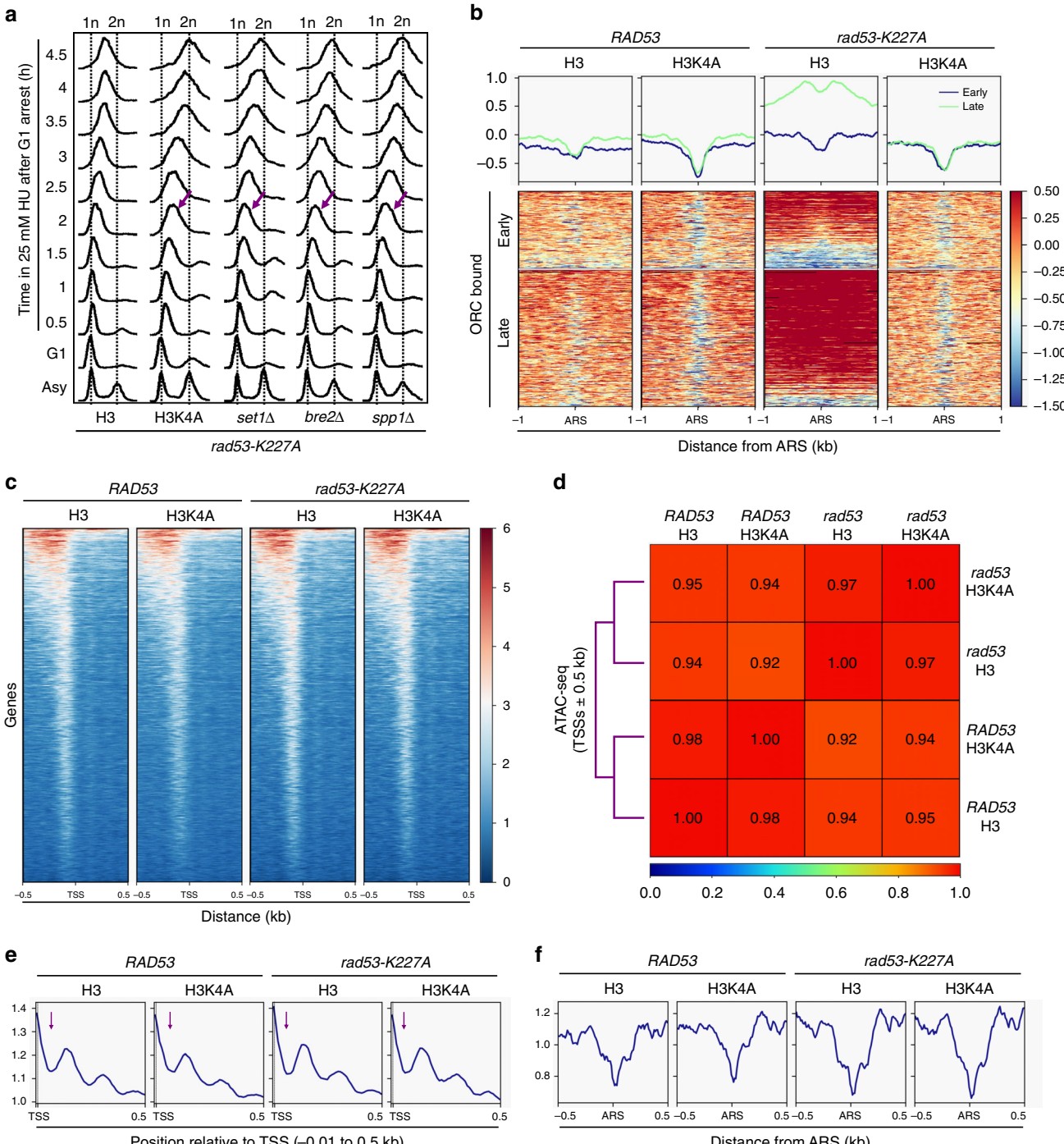

**Fig. 3 H3K4 methylation causes fork stalling in *rad53* mutants following replication stress without altering chromatin structure. a** Cells with H3, H3K4A, *set1Δ*, *bre2Δ*, and *spp1Δ* on the *rad53-K227A* background were arrested in G1 and released into 25 mM of HU for 2 h. The cellular DNA content was determined by FACS at the indicated time-points. Purple slanted arrows indicate faster replication progression in mutant cells compared to H3 cells. **b** The graphs show DNA Pol2 binding profiles centered on ORC-bound early or late *ARSs* in *RAD53*-H3, *RAD53*-H3K4A, *rad53-K227A*-H3, and *rad53-K227A*-H3K4A cells. **c** ATAC-seq signals on TSSs ± 0.5 kb regions after G1-release in 25 mM HU. **d** Spearman's rank correlation coefficient of ATAC-seq signals according **c**. **e** ATAC-seq signals on TSS regions (−0.1 to 0.5 kb). **f** ATAC-seq signals on ORC-bound origins.

in *rad53*-mutant cells. Furthermore, transcription and H3K4 methylation, which are coupled processes[27], may make parallel contributions to aberrant replication events in *rad53* mutants exposed to replication stress.

We next addressed whether the orientation of the transcription-replication collisions could impact the appearance of replication defects in cells with defective Rad53. A new strain

was generated in which the *pMET25-LRE1* was reoriented, such that it could be transcribed co-directionally (CD) with the replication fork movement originating from *ARS305* (Fig. 4g, CD-*pMET25-LRE1*). Interestingly, the activation of CD-*pMET25-LRE1* promoted accumulation of large Y arcs within *rad53* cells in 25 mM HU, indicating that the forks emanating from *ARS305* were routinely stalled (Fig. 4g, blue box); meanwhile, the

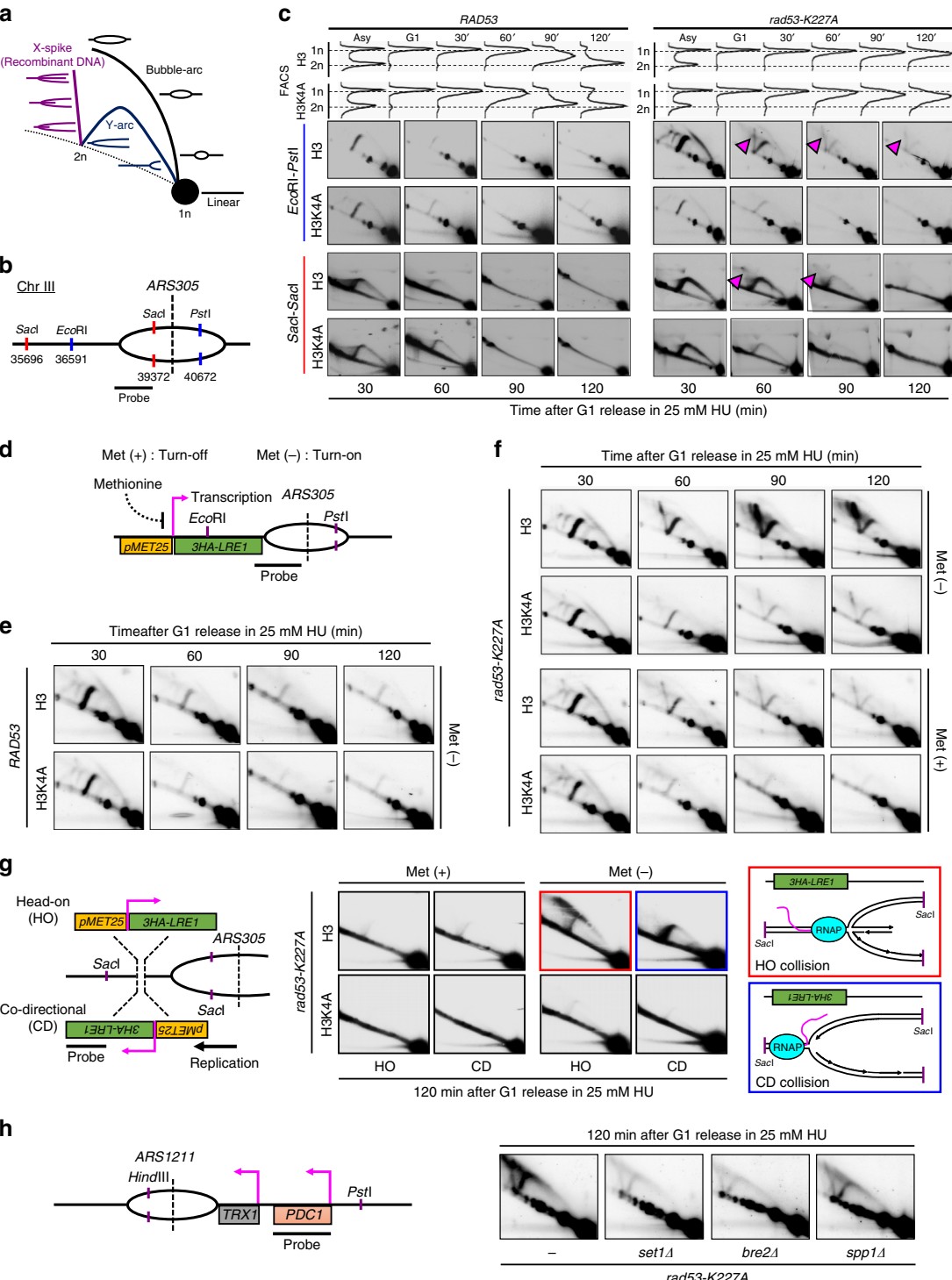

activation of head-on (HO) *pMET25-LRE1* promoted fork reversal (Fig. 4g, red box), which may be explained by the previous finding that head-on conflicts are much more detrimental to genome stability[10,11,50,51]. These two distinct types of impeded replication progression were both prevented in *rad53*-H3K4A cells, demonstrating that H3K4 methylation aggravates replication fork impediments in *rad53* mutants independent of the orientation of collisions.

In addition to studying the role of H3K4me on replication fork progression in the context of an artificially induced TRC, we examined an endogenous TRC. For this experiment, we analyzed the replication forks emanating from an early-firing origin,

*ARS1211*, which is naturally positioned next to *PDC1*, a highly transcribed ORF, with an orientation that facilitates head-on collisions between replication and transcription machinery. In *rad53* mutants with 2 h HU treatment, *ARS1211* exhibited high levels of large Y arcs and aberrant replication intermediates, indicating the movement of replication forks was stalled, generating reversed forks. These aberrant fork signals were dramatically reduced in *set1Δ rad53* or *bre2Δ rad53* cells and somewhat lower in *spp1Δ rad53* mutants (Fig. 4h). Together, these results demonstrated that H3K4 methylation impeded fork progression in HU-treated *rad53* mutants in a methylation level-dependent manner.

**Fig. 4 Loss of H3K4 methylation and transcription progression prevents defective replication fork formation in *rad53* mutants. a** Schematic representation of the signal profile for 2D gel analyses. **b** A schematic representation of *ARS305* for 2D gel analysis is shown. **c** 2D gel analysis after in vivo psoralen crosslinking of the *Eco*RI-*Pst*I-digested (blue) or *Sac*I-digested (red) fragments of the *ARS305* region in *RAD53*-H3, *RAD53*-H3K4A, *rad53*-K227A-H3, and *rad53*-K227A-H3K4A cells released from G1 into 25 mM HU at the indicated time-points. FACS profiles show the cellular DNA content during the experiments. The purple arrows indicate spike signals corresponding to aberrant DNA intermediates that accumulated in HU in *rad53*-K227A mutants. **d** A schematic representation of the inducible *MET25* promoter driving the *3HA-LRE1* gene with a head-on orientation to *ARS305*. Met (+) indicates S. C. medium, which suppresses *3HA-LRE1* by inactivating the *MET25* promoter. Met (−) indicates S. C. minus methionine medium, which turns on *3HA-LRE1* by the *MET25* promoter. **e** 2D gel analysis of the *Eco*RI-*Pst*I-digested fragments on *ARS305* region in *RAD53*-H3 and *RAD53*-H3K4A cells released from G1 into 25 mM HU at the indicated time-points in Met (−) medium. **f** As in **e**, 2D gel analysis of *rad53*-K227A-H3 and *rad53*-K227A-H3K4A cells in Met (−) or Met (+) medium. **g** 2D analysis of *Sac*I-digested replication intermediates from *rad53*-K227A-H3 and *rad53*-K227A-H3K4A cells with either HO (head on)-TRC or CD (co-directional)-TRC between *3HA-LRE1* transcription and fork originated from *ARS305* following G1 synchronization and subsequent release into 25 mM HU for 2 h. Met (−) and Met (+) as in (**a–c**). A schematic representation of putative replication fork intermediates represented by the 2D-gel signal shown in HO-TRC (red box) and CD-TRC (blue box). **h** 2D gel analysis of *Hind*III-*Pst*I-digested DNA fragments from *rad53*-K227A and *set1Δ*, *bre2Δ*, or *spp1Δ* on the *rad53*-K227A background to monitor replication intermediates at *ARS1211* after G1 release into HU treatment for 2 h.

## H3K4me promotes aberrant replication forks in *rad53* mutants

The deposition of H3K4 methylation on genes is coupled with transcriptional activity[27]; however, our results suggest that transcription and H3K4 methylation may work in parallel to impede fork progression in HU-treated *rad53* mutants (Fig. 4f). To address whether H3K4me directly promotes aberrant events at replication forks, we first kept *pMET25-LRE1* gene in its active state to allow H3K4me deposition (Supplementary Fig. 4c). Then, we repressed its transcription just before replication forks passed through (Fig. 5a, On → Off). To preserve the H3K4me that was deposited during the active transcription phase, we deleted *JHD2*, which encodes a H3K4 demethylase that is presumed to erase H3K4me3 during cell cycle progression[29,52]. The deletion of *JHD2* had no effect on *rad53* HU-sensitivity (Fig. 5b), and the total levels of H3K4 methylation under normal growth conditions were also not affected (Fig. 5c)[53]. We found that following the "On → Off" medium switching procedure (Fig. 5a), the expression of *pMET25-LRE1* was largely suppressed but H3K4me3 remained abundant (Fig. 5d, e), and fork reversal was clearly detected in HU-treated *jhd2Δ rad53* double mutants (Fig. 5f, purple arrows). On the contrary, aberrant replication events were not detected when *pMET25-LRE1* was suppressed at all times (Off), and H3K4me3 was present at relatively lower levels (Fig. 5e, f). Thus, we conclude that the restraint of replication progression by H3K4 methylation in *rad53* mutants upon HU-stress is independent of RNA pol II transcription.

## A speed bump model

While the role of H3K4me gradient in regulating transcription is still under debate[26,27], we propose that high levels of H3K4 methylation deposited by high transcriptional activity result in a large cushioning effect on fork progression to protect against transcription-replication collisions. We refer to this idea as the "speed bump model" (Fig. 6a). We reasoned that under HU-induced replication stress, the activation of Rad53 checkpoint kinase maintains the stability of replisomes[54], which may be expected to proceed independently of transcriptional activity. Thus, the differential patterns of H3K4me provide transcription-deposited "bumps" to decelerate fork movement at genomic regions where transcriptional activity is high. These transcription-regulated "speed bumps" could therefore specifically protect potential zones of conflict, where transcription and replication machineries are likely to meet, while allowing faster replication in areas with no or low transcriptional activity. In the absence of checkpoint activity, such as in *rad53*-mutant cells, H3K4 methylation becomes an impediment to replication and destabilizes the replisome.

## H3K4me is a molecular decelerator for fork progression

To test the "speed bump model", we performed FACS analysis and showed that bulk DNA synthesis in a single cell cycle was faster in H3K4A mutant cells compared to WT (H3) in both *RAD53* and *rad53* mutant backgrounds with or without 200 mM HU (Fig. 6b). Using 2D gels to monitor replication fork progression, we probed windows that extended ~20 kb from *ARS305* and *ARS1211* to track replication forks movement in WT and mutant strains. Cells were released from G1-synchronization into 200 mM HU for 60 min; under these conditions, active Rad53 kinase is required to protect stalled forks[55]. Clear and solid Y-arc signals were observed in the probed regions near *ARS305LL* and *ARS1211R* in *RAD53*-H3K4A cells, but were less obvious in *RAD53*-H3 cells (Fig. 6c, red arrows), suggesting the replication forks in *RAD53*-H3K4A cells migrated faster. In *rad53* cells, large Y arcs and coned signals accumulated in the regions near *ARS305* and *ARS1211* due to fork collapse, while bubble-arc structures were found in the late origin (*ARS1212*) due to uncontrolled late origin firing[56]. However, the ablation of H3K4 methylation in *rad53* cells allowed replication forks to migrate further in the windows near *ARS305L*, *ARS305LL* and *ARS1211R*, though aberrant fork structures persisted (Fig. 6c, blue arrows). In sum, we conclude that chromatin templates modified by H3K4 methylation may serve as a molecular decelerator of replication fork progression, consistent with our "speed bump model".

## H3K4me defenses the TRC-mediated instability under HU stress

The recent finding that accelerated fork progression leads to genome instability[57] is reminiscent of our "speed bump model" and prompted us to propose that histone H3K4me is crucial to protect cells from HU-induced replication perturbations. This hypothesis was supported by the observation that deletions of Set1C subunits or H3K4 mutation increase sensitivity to high-dose HU, eventually leading to cell death (Fig. 7a)[31]. We then proposed that H3K4 methylation may functionally couple with the S-phase checkpoint to protect genome stability. To test this hypothesis, we measured *CAN1* mutation rate in WT and mutant strains. The *CAN1* gene is widely used as a mutation reporter gene in yeast and is thought to reflect the stability of the entire genome[58]. The spontaneous mutation rate of the *CAN1* gene in *rad53*-H3K4 cells appeared to be slightly elevated, although the effect was not statistically significantly (Fig. 7d, left panel). We then suspected that the mutation rate may correlate with TRCs. Methylated H3K4 is more abundant on highly transcribed genes where TRCs are more likely to occur; therefore, we speculated that the genome protection by H3K4me may be linked to transcription frequency. To test this possibility, we utilized a strong *TEF1* promoter to drive the *CAN1* gene. We observed elevated and extended H3K4me3 on the *pTEF-CAN1* gene (Fig. 7b, c), in agreement with the notion that H3K4me patterns are dependent on frequency of transcription[27]. Strikingly, the spontaneous

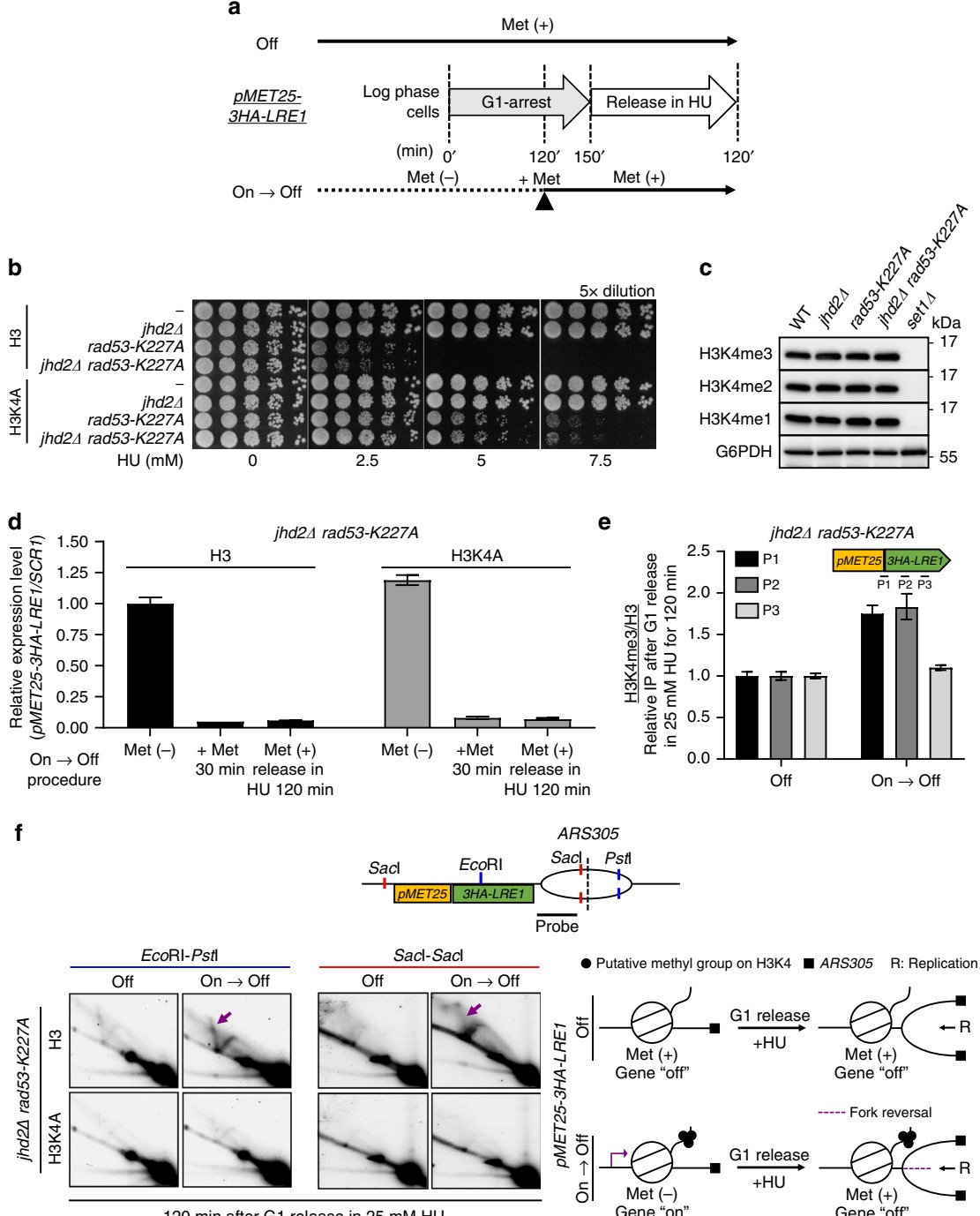

**Fig. 5 H3K4 methylation promotes defective replication fork formation in *rad53* mutants. a** A schematic representation of the experimental procedure to examine the effect of H3K4me on HU-treated *rad53* mutants. **b** HU sensitivity was determined for *JHD2* and *jhd2Δ* mutants on the *RAD53* or *rad53-K227A* backgrounds with wild-type H3 or H3K4A mutation. **c** Immunoblot showing H3K4me1/2/3 levels in *jhd2Δ* strains. **d** Normalized *pMET25-3HA-LRE1* expression in *jhd2Δ rad53-K227A* with wild-type H3 or H3K4A mutation **e** H3K4me3 deposition on indicated regions of *pMET25-3HA-LRE1* in in *jhd2Δ rad53-K227A* with wild-type H3 after following the procedure show in **a**; all error bars represent ± SEM (*n* = 3 technical replicates). **f** 2D gel analysis of *Eco*RI-*Pst*I-digested and *Sac*I-*Sac*I-digested fragments of the *ARS305* region in *jhd2Δ rad53-K227A* with wild-type H3 or H3K4A mutated cells. In brief, cells were G1-released in 25 mM HU with suppressed *pMET25-3HA-LRE1* transcription but with low (Off) or high (On → Off) levels of residual H3K4me. The purple arrows indicate spike signals corresponding to aberrant DNA intermediates that accumulated in HU treated *rad53-K227A* mutant with high level of H3K4me deposition.

mutation rates of *pTEF-CAN1* in all our strains were increased at least 5-fold over the native *CAN1* gene (Fig. 7d), consistent with the idea that high levels of transcription increased the spontaneous mutation rate as a result of increased TRCs[5,45,59]. Notably, the *RAD53* mutation and loss of H3K4 methylation together

significantly elevated the *pTEF-CAN1* mutation rate compared to cells with the wild-type *RAD53* and H3, or cells with wild-type *RAD53* and H3K4A (Fig. 7d). Unlike *rad53*-H3 cells, *rad53*-H3K4A cells can survive in low-dose (5 mM) HU (Fig. 1a), however, we found the survivors had significantly more

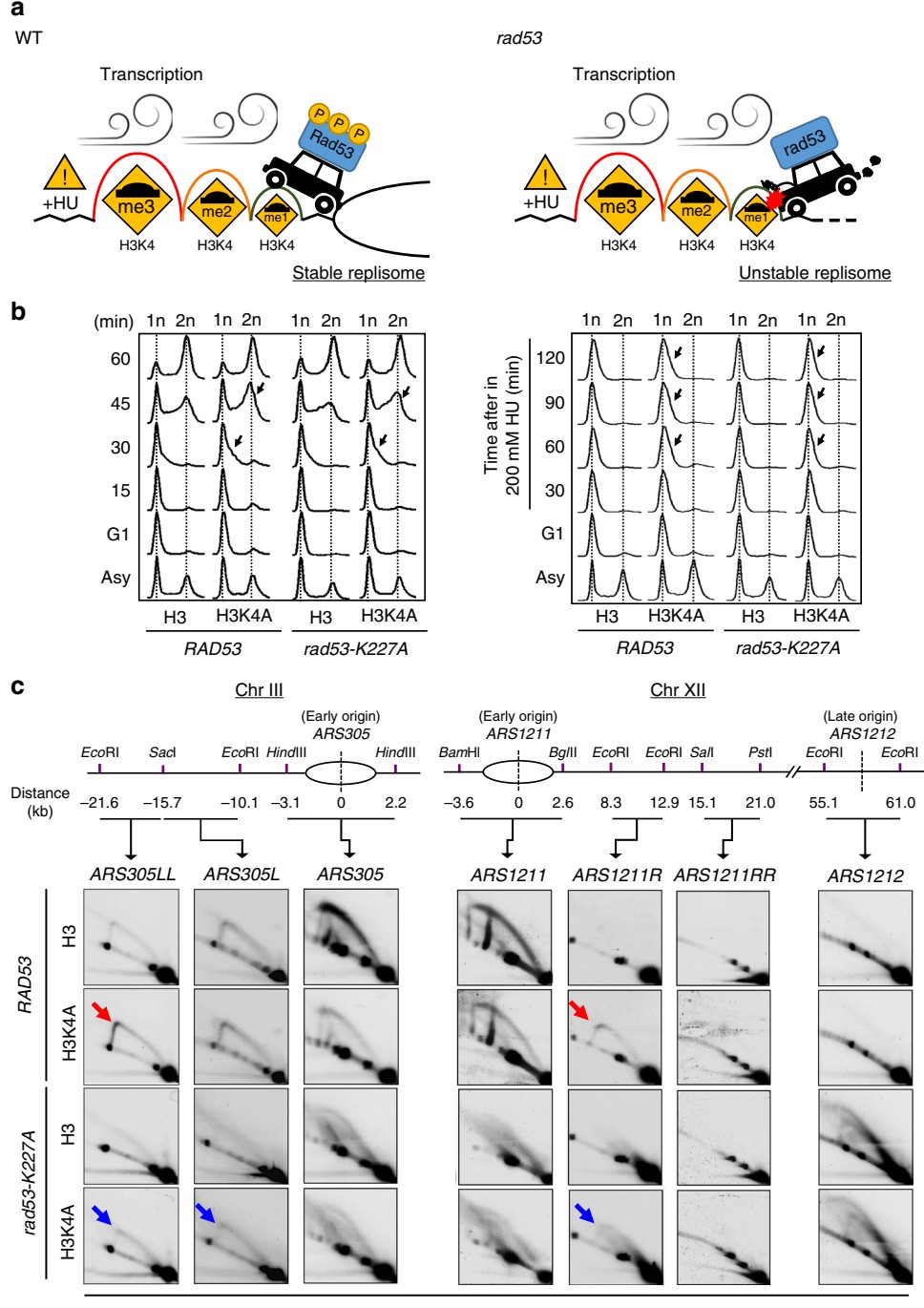

**Fig. 6 H3K4 methylation deposited by transcription decelerates replication forks. a** A graphic depiction of the "speed bump model". Sites of H3K4 methylation deposited by transcriptional activity act as "speed bumps" for active replication forks (represented by the black car), while activation of Rad53 checkpoint kinase maintains the stability of replisomes during replication stress. In the absence of checkpoint activity, such as in *rad53*-mutant cells, H3K4 methylation becomes an impediment to replication and destabilizes the replisome. The red explosion drawing indicates the stalling of replication forks. **b** FACS profiles of *RAD53*-H3, *RAD53*-H3K4A, *rad53-K227A*-H3, and *rad53-K227A*-H3K4A cells released from G1 into rich medium without (left) or with 200 mM HU (right) at indicated time points. Black arrows indicate faster S-phase progression or DNA duplication in H3K4A cells compared to H3 cells. **c** 2D gel analysis of *RAD53*-H3, *RAD53*-H3K4A, *rad53-K227A*-H3, and *rad53-K227A*-H3K4A cells. Replication intermediates from indicated regions of chromosomes III and XII from *ARS305* and *ASR1211* were identified in the indicated strains following G1 synchronization and release into 200 mM HU for 1 h. The arrows mark Y-arc signals that represent duplicating replication forks in *RAD53*-H3K4A and *rad53K227A*-H3K4A cells. Bubble-arc signals at *ARS1212* indicate the late origin was repressed in *RAD53* and activated in *rad53-K227A* cells.

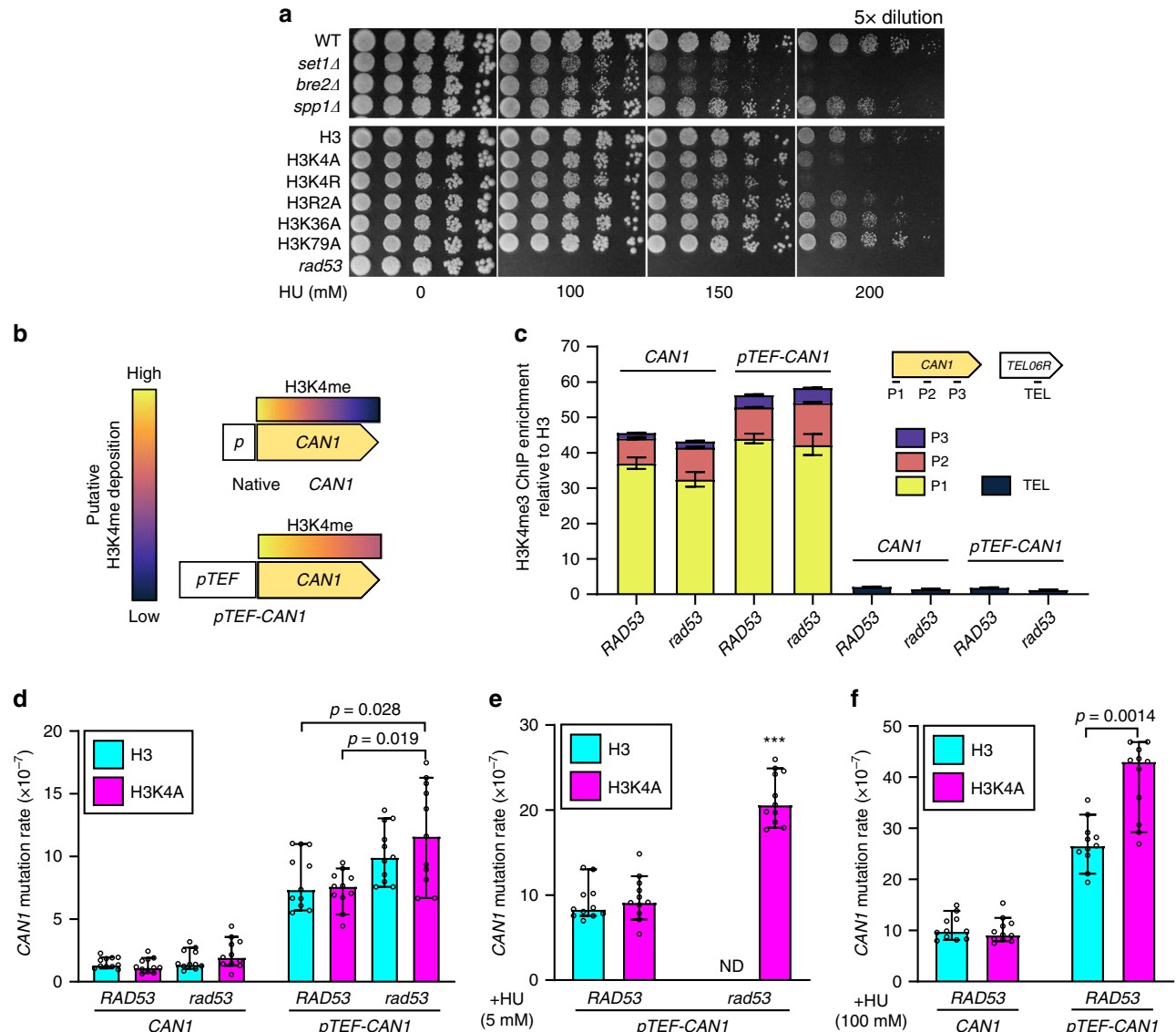

**Fig. 7 H3K4 methylation defends the genome from TRCs-mediated instability under replication stress. a** HU sensitivity was determined for WT and isogeneic *set1Δ*, *bre2Δ*, and *spp1Δ* strains, as well as the H3 strain and isogeneic H3K4A, H3R2A, H3K36A and H3K79A mutants. **b** Putative elevated H3K4me and **c** actual H3K4me3 after *pTEF* promoter was introduced into the *CAN1* gene; error bars represent ± SEM (*n* = 3 technical replicates). Mutation rates of **d** native *CAN1* (left panel) and *pTEF-CAN1* (right panel) were measured by the fluctuation test (see Methods section) in *RAD53*-H3, *RAD53*-H3K4A, *rad53*-H3 and *rad53*-H3K4A cells. *rad53* refers to *rad53-K227A* mutant. As in **d**, cells were treated by **e** 5 mM and **f** 100 mM HU. Data are the median with 95% confidence intervals (*n* = 11 independent samples). Statistical significance was determined by Mann–Whitney *U*-test; *p* < 0.05; asterisks represent *p* < 0.001 compared with *RAD53*-H3 or *RAD53*-H3K4A; ND, not determined.

mutations at *pTEF-CAN1* (Fig.7e). This result suggests that the ablation of H3K4 methylation promotes the survival of *rad53* cells in HU by restoring replication fork progression in a short term, but likely with a cost to genome integrity in a long term. Intriguingly, sub-lethal HU treatment significantly elevated the mutation rate of *pTEF-CAN1* in WT (H3) cells, and the mutation rate was further increased in H3K4A cells (Fig. 7f). These data suggested that H3K4 methylation works in coordination with the Rad53 checkpoint to protect genomic stability, especially at highly transcribed genomic regions where this histone mark is actively deposited. Taken together, our results established that H3K4 methylation protects genome integrity by slowing down fork migration, especially on highly transcribed regions, to mitigate TRCs under sublethal replication stress.

## Discussion

Oncogenic stress drives the early stages of cancer development[60] and has been shown to preferentially target certain genomic regions, such as CFSs and perhaps ERFSs[15,61]. In yeast, HU-induced replication fork collapse has been observed preferentially around genomic loci where early-firing replication origins may encounter transcription[62], similar to ERFSs in the human genome[15]. Here, our analyses suggest that H3K4 methylation works together with the S-phase checkpoint kinase, Rad53, playing a key role in reducing collisions between replication and transcriptional machinery to protect genome integrity. We propose that H3K4 methylation mitigates TRCs by slowing down the replicon as it passes through highly expressed ORFs, where this histone mark is abundant, and this action promotes faithful replication (Fig. 8a).

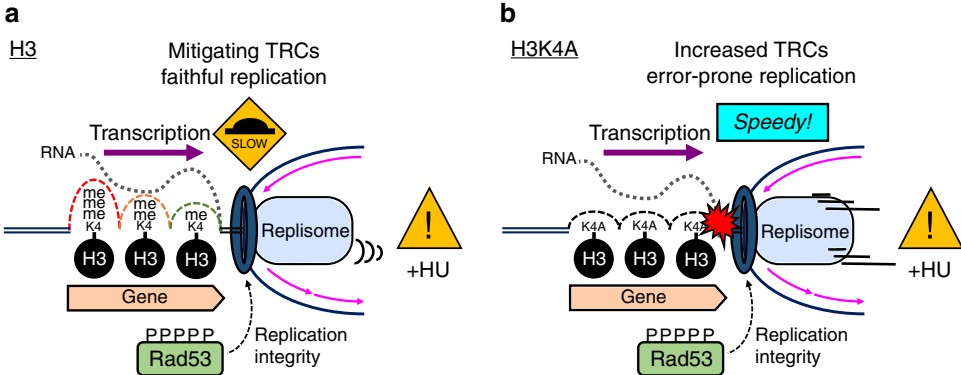

**Fig. 8 Model for the effect of H3K4 methylation in relieving transcription-replication conflicts. a** Upon HU-stress, the S-phase checkpoint kinase Rad53 is activated to stabilize replication fork integrity; meanwhile, transcription-deposited H3K4 methylation acts as a series of speed bumps to decelerate fork progression, relieving transcription-replication conflicts to reduce DNA damage at highly transcribed regions. **b** Conversely, without H3K4 methylation, rapid fork movement aggravates transcription-replication conflicts (red explosion drawing) and leads to error-prone replication and genome instability.

Several lines of evidence support this conclusion: (1) our epistasis studies suggest that H3K4me and Rad53 function in the same pathway in the S-phase checkpoint; (2) H3K4 methylation causes fork stalling and leads to abnormal replication fork formation in *rad53* mutants following replication stress; (3) H3K4 methylation decelerates fork progression in early replication origins; (4) H3K4 methylation prevents mutagenic events induced by TRCs.

In eukaryotic cells, multiple origins of replication are coordinately activated during S phase, and the velocity of replication forks is modulated such that the entire genome can be replicated in pace with the cell cycle[63]. In addition, transcriptional activity varies among genomic regions. It is known that replication is paused by high transcriptional activity[8], and S-phase checkpoint activity is required for the replication apparatus to pass through highly transcribed genes[2]. However, it is unclear how the checkpoint modulates replication forks as they traverse genomic regions with different levels of transcription. The finding that high-speed fork progression induces replication stress and genome instability[57] resembles the effect of losing H3K4 methylation, wherein H3K4A cells also exhibit speedy fork progression and error-prone replication (Fig. 8b). Thus, the detrimental effect of losing H3K4me in normal cells under replication stress provides a compelling explanation for the evolutionary conservation of H3K4 methylation at actively transcribed loci. Recent exome-sequencing studies have shown that dysregulation of MLL family proteins has impacts well beyond those described for rare cancers like MLL1-rearranged mixed-lineage leukemia[64,65]. In fact, MLL family mutations are among the most frequent alterations in human cancer[66] and have been associated with some of the most common and deadly solid tumors, such as lung[67] and colon carcinomas[68]. Recent observations also indicate that this highly conserved histone modification may play an important role in DNA damage response[30,69,70] and DNA replication fork stabilization under stress[32,71]. However, it is unclear how this histone mark, which is enriched by transcriptional activity, contributes to maintaining genome stability. Our results suggest a molecular mechanism by which H3K4me contributes to mitigating TRCs in replication stress, thus promoting genome stability. Moreover, our model provides a plausible reason for the positive correlation between transcriptional activity and the gradient patterning of H3K4me (Fig. 8a).

Unlike acetylation, neutrally charged modifications, such as histone methylation, are unlikely to directly perturb chromatin structure since these modifications are small and do not alter the charge of histones[72]. Therefore, one possible mechanism for H3K4 methylation to regulate fork progression is through recruitment of some protein complex to the nucleosome, which may form a physical barrier or limit histone eviction efficiency. H3K4me3 is recognized by a PHD finger within the ING family of proteins (ING1-5)[73]. The ING proteins in turn recruit additional chromatin modifiers such as histone acetyltransferase complexes (HATs) and histone deacetylase complexes (HDACs). For example, ING2 tethers the repressive mSin3a-HDAC1 complex to active proliferation-specific genes following DNA damage[74]. H3K4me3 is also bound by the tandem chromodomains within CHD1, an ATP-dependent remodeling enzyme capable of repositioning nucleosomes[75], and by the tandem Tudor domains within JMJD2A, a histone demethylase[76]. In yeast, it was previously reported that H3K4 methylation readers include Chd1, Isw1, Yng1, Yng2, Pho23, and Set3[77,78]. Notably, H3K4me3 may recruit Yng1, which binds via its PHD finger[79]. This protein then associates with NuA3 HAT, leading to histone acetylation around the promoter and subsequent gene activation. H3K4me3 may also recruit the Rpd3L complex (with Pho23 as the H3K4me3 reader), which promotes histone deacetylation and represses transcription[80]. Thus, the two opposing enzyme activities are both associated with H3K4me3 at promoters. Furthermore, another level of complexity is added by the fact that the Set3 complex HDAC can be recruited by H3K4me2[81]. Even though the H3K4 methylation gradient is known to be the result of transcription[26,27], it is not clear whether and how these complex downstream mechanisms contribute to chromatin dynamics or if chromatin dynamics mediate the deceleration of replication fork migration. Our current study demonstrates that H3K4me-enriched genomic regions require more time for traversal of the replisome. As such, the frequency of conflicts is reduced, providing a safeguard against TRCs between transcription and replication machinery. It will be interesting to further investigate the molecular mechanisms of how H3K4 methylation attenuates mutations at highly transcribed regions upon replication stress. Possible mechanisms may involve specific interactions between the replisome complex and H3K4me-decorated chromatin templates that improve DNA synthesis accuracy either by fine-tuning the biochemical activities of replication machinery, by promoting transcription-coupled repair[82,83], or by reducing chromatin torsional stress during TRCs[84,85].

Synthetic rescue phenotypes often provide powerful clues to mechanisms underlying cellular processes; hence, the identities of *rad53* HU-sensitivity suppressors shed light onto how Rad53 checkpoint and replication stress response function. Several suppressors of *rad53*-HU-sensitivity have been identified. Deletion of either the G1/S or G2/M factor delays cell cycle transition[86]; THO, TREX-2 components alter nascent RNA exportation[17]; and Rrm3 and Pif1 DNA helicases[34] allow *rad53*

mutants to survive HU stress. This broad nature of the suppressors likely reflects the complexity of the mechanisms by which Rad53 functions. Nonetheless, the interaction between H3K4 methylation and *RAD53* are genetically independent of other potential TRC-regulating factors (Supplementary Fig. 2). The suppression of HU sensitivity by ablation of H3K4 methylation provides a clue to unravel the functional orchestration of the Rad53 checkpoint kinase and histone modifications.

A number of different mechanisms have been put forward to explain how the multifaceted S-phase checkpoint contributes to genome protection[87]. Following depletion of dNTPs, this highly conserved checkpoint kinase pathway preserves the functionality and structure of stalled DNA replication forks and prevents chromosome fragmentation. In this study, we have uncovered a function of H3K4 methylation by showing that the coordination between Rad53 kinase and H3K4 methylation contributes to genome stability in response to HU treatment, particularly in highly expressed regions. We propose the "speed bump model" to illustrate a previously unknown function of the canonical H3K4 methylation gradient on active genes to prevent collision between transcription and replication machineries. Our finding thus provides a perspective to further understand the role of H3K4 methylation in genome instability and cancer.

## Methods

**Yeast strain construction.** All yeast strains with indicated H3 mutations were developed from the MSY421 background, a strain with all copies of the H3–H4 genes deleted and extrachromosomal H3–H4 genes to compensate for the deletions [pMS329 (*CEN4 ARS1 URA3 HHT1 HHF1*)]. All H3 mutants were generated using pMS337 [pMS337 (*CEN4 ARS1 LEU2 HHT1 HHF1*)] by site-directed mutagenesis. To generate the mutant stains with indicated H3 mutations, the H3-H4 genes in pMS329 were replaced by wild-type or mutated H3–H4 genes in the pMS337 plasmid using standard plasmid shuffling methods. To construct the inducible TRC yeast strains used in the 2D gel assay, we integrated *pMET25* and fused a 3× HA tag on the *LRE1* gene using a PCR fragment from pYM-N36. The *pMET25-3HA-LRE1* allele was cloned and integrated to replace the *lre1::URA3* allele (MSY421 *bar1:: HIS3 lre1::URA3*), generating marker-free HO- and CD- *pMET25-3HA-LRE1* strains. For yeast strains used in mutation assay, we integrated *pTEF* to overexpress *CAN1* gene using a PCR fragment from pYM-N18. The details of strains, sequences of primers used in site-directed mutagenesis and indicated strains construction are listed in Supplementary Tables 1 and 2.

**Yeast spotting assay.** Cells were resuspended at $2.5 \times 10^6$/ml and subjected to 5-fold serial dilution in distilled water, and an aliquot of 2.5 µl of each dilution was spotted onto the indicated agar plates. Growth was recorded after incubation at 30 °C for 3 days.

**Cell viability assay.** Log-phased cells were G1 arrested by 100 ng/ml α-factor for 2.5 h and then released into a medium with 25 mM HU. Aliquots were collected from each culture at the indicated time-point, plated onto YPD plates, and allowed to grow at 30 °C for 3 days. Cell viability was estimated based on colony forming unit (CFU) counts, and data were normalized by each control group (time-point at 0), respectively.

**Western blotting.** Total cell lysates were prepared by TCA protein extraction method. Cells were resuspended in TCA buffer (1.85 M NaOH and 7.4% β-mercaptoethanol) for 10 min and then mixed with equal volume of 20% trichloroacetic acid to precipitate lysates. Cell lysates were dissolved in 0.1% NaOH and resolved using SDS-PAGE, then transferred onto PVDF membranes (Immobilon®-P, Millipore); membranes were then incubated with the indicated antibodies. The following antibodies were used in this study (dilutions used, source and catalog numbers): α-H2A S129-P (1:2000, Abcam, ab15083), α-H3K4me3 (1:10,000, Abcam, ab8580), α-H3K4me2 (1:5000 Abcam, ab7766), α-H3K4me1 (1:2000, Abcam, ab8895), α-HA (1:5000, Roche, 11867423001) and α-G6PDH (1:20,000, Sigma, A9521). Signals were detected using the ECL detection reagent (Millipore Immobilon™) with UVP BioSpectrum® imaging system. Uncropped blots can be found in Source data.

**Flow cytometry analysis.** Cells (with *bar1Δ*) were synchronized at the onset of log-phase by 100 ng/ml α-factor for 2.5 h and were released from G1 into experimental conditions. Collected cells were fixed in 70% ethanol and were then treated with 1 mg/ml RNase A at 37 °C in 50 mM Tris-HCl (pH 8.0). Fixed cells were washed and treated with 2 µl protease K (>800 units/ml) in 100 µl 30 mM Tris-HCl. Next, the cells were stained with SYBR™ Green I Stain (1:3000,

Invitrogen, S7563) in 50 mM Tris-HCl (pH 8.0) overnight at 4 °C and were analyzed using a flow cytometer (BD FACSCanto™ II). A total of 10,000 gated cell signals were analyzed for each time-point using FlowJo software.

**Chromatin immunoprecipitation (ChIP)—sequencing.** Log-phase cells were treated with 25 mM HU for 2 h. Cells were fixed with formaldehyde solution (5 mM HEPES [pH 7.6], 0.1 mM EDTA, 10 mM NaCl and 1.1% formaldehyde) for 20 min and then quenched with 125 mM glycine for 5 min. Cell pellets were washed following resuspended in 10 ml cold TBS (20 mM Tris [pH 7.6] and 150 mM NaCl). Pellets were sonicated for two cycles of 30-sec-on and 30-sec-off, followed by one cylce of 10 min in 700 µl FA buffer (50 mM HEPES-KOH [pH 7.6], 150 mM NaCl, 1 mM EDTA, 1%Triton X-100, 0.1% sodium deoxycholate and 1× protease inhibitor cocktail). The DNA content of cell lysates was measured using the Qubit fluorescence spectrophotometry system, and 1 µg of DNA from each sample was used for ChIP. Antibody was conjugated to Dynabeads™ (Invitrogen) in sodium phosphate buffer (40 mM $NaH_2PO_4$ and 60 mM $Na_2HPO_4$) and washed twice with sodium phosphate buffer with 0.01% Tween-20 and three times with PBS-T (PBS with 0.01% Tween-20). The antibody-conjugated beads were resuspended in 30 µl of PBS-T for IP assay. One hundred and fifty microliter of sonicated chromatin were mixed with antibody-conjugated beads, incubated at room temperature for 1.5 h. Chromatin-bound beads were washed by FA buffer for three times, then by FA-HS buffer (FA buffer with 500 mM NaCl) for two times following by RIPA buffer (10 mM Tris [pH 8.0], 0.25 M LiCl, 0.5%NP-40, 0.5% sodium deoxychalate, 1 mM EDTA and 1× protease inhibitor cocktail) for one time. Bound-chromatins were eluted by stop buffer (20 mM Tris [pH 8.0], 100 mM NaCl, 20 mM EDTA and 1% SDS) at 75 °C for 15 min. The eluates were incubated at 75 °C for overnight prior to 2 µl proteinase K (10 mg/ml) treatment at 50 °C for 6 h and 2 µl RNase A (10 mg/ml) at 42 °C for 2 h. DNA were purified by using MinElute® PCR Purification Kit (QIAGEN) in 12 µl elution buffer. For Rpb3-ChIP, 5 µl of anti-Rpb3 antibody (Biolegend, 665003) was conjugated to 20 µl Dynabeads™ M-280 sheep anti-mouse IgG beads (Invitrogen, 11201D). For DNA Pol2-ChIP, the target protein was epitope-tagged with 3× FLAG and immune-precipitated using 5 µl of anti-FLAG antibody (Sigma, F3165) and conjugated to 20 µl Dynabeads™ Protein G beads (Invitrogen, 10004D). Libraries were prepared using the Ovation® Ultralow v2 kit (NuGEN, catalog #0344). Single-end sequencing was completed on the HiSeq™ 2500 platform (Illumina) in rapid run mode at the Fred Hutchinson Cancer Research Center genomics core facility.

**ATAC-seq.** G1-arrested cells were released in 25 mM HU for 2 h; total $5 \times 10^5$ cells were spheroblasted in 400 µl sorbitol buffer (1.4 M sorbitol, 40 mM HEPES-KOH [pH 7.5], 0.5 mM $MgCl_2$) with 0.5 mg/ml Zymolyase®-100T (Nacalai Tesque, Inc.) and 10 mM DTT for 15 min at 30 °C prior to incubation with reaction buffer (TD), Tn5 Transposase (TDE1) from Nextera™ kit and additional 0.01% digitonin in total volume of 50 µl at 37 °C for 30 min. Transposed DNA was purified by MinElute™ PCR Purification Kit (QIAGEN) and then amplified by PCR with custom primers from Nextera™ DNA CD Indexes kit. Amplified libraries were then purified by using MinElute™ PCR Purification Kit (QIAGEN). Paired-end sequencing was completed on the NextSeq™ 500 (Illumina) with MID output (150 cycles) at the Institute of Molecular Biology, Academia Sinica.

**ChIP-seq and ATAC-seq data analysis.** For the ChIP-seq analysis, FASTQ files from biological and technical replicates for each sample were merged. Reads were aligned to the sacCer3 reference genome (release R64-2-1) using Bowtie2 version 2.2.5 in "very-sensitive" mode[88]. Aligned reads were filtered and indexed using SAMtools. Reads were adjusted so that the genome average was set at 1-fold enrichment. Then, data from immunoprecipitated samples were divided by data from input samples. Peak calling was completed using the "callpeak" command in MACS v2 software, and shared peaks between samples were determined using the MACS "bdgdiff" command. Heatmaps were generated using deepTools. Transcription frequency was calculated as the total signal in a range from the TSS to TSS + 1000 bp of each yeast gene adapted from the results of NET-seq[89]. For ATAC-seq analysis, sequencing reads were aligned and processed with similar method described in ChIP-seq data, except using the default "sensitive" mode and -X 2000 for limiting the maximum fragment length to 2000. Aligned reads were indexed and filtered by Sambamba to remove reads that were aligned to chrM, with duplicates, not proper paired and low mapping quality (<30). HMMRATAC was used for peaking calling with the parameter "--trim 1" to remove the 3 N signal track. We further defined reads with fragment length shorter than 58 as nucleosome-free and reads with fragment length range in 1 standard deviation of the 1 N signal tract estimated by HMMRATAC as nucleosome-enriched. Only the fragments lower than 58 were used to define chromatin accessibility. Heatmaps were generated using deepTools.

**ChIP-qPCR.** Cells were harvested according to the indicated culture conditions or treatments, followed by ChIP procedures. Sonicated chromatin lysates from cells (≈$1.8 \times 10^8$) were used for ChIP. For H3-ChIP and H3K4me3-ChIP, 2.5 µl of anti-H3 antibody (Abcam, ab1791) and 1 µl of anti H3K4me3 antibody (Abcam, ab8580) were conjugated to 20 µl Protein G Mag Sepharose™ (GE Healthcare), respectively. qPCR was performed using LightCycler® 480 Instrument II (Roche). The sequences of primers for qPCR are listed in Supplementary Table 2.

**Gene expression**. RNA was extracted from cells with hot acid-phenol. Four thousand nanogram of total RNA was reverse transcribed into cDNA using random primers and the SMART® MMLV Reverse Transcriptase (Takara). cDNA was analyzed by real-time qPCR using KAPA SYBR® FAST qPCR Kit (Kapa Biosystems) and StepOne Plus™ system (ABI). The sequences of primers used to amplify each gene are listed in Supplementary Table 2.

**2D gel electrophoresis**. All strains for the 2D gel analysis were *BAR1*-deleted. Log-phase cells ($OD_{600} \approx 0.5$) were synchronized at G1 by α-factor (100 ng/ml) for 2.5 h in appropriate medium. Cells were then washed and released in medium at 30 °C with the indicated dosage of HU. $\approx 3 \times 10^{10}$ cells were harvested at indicated times and terminated immediately using 0.1% (w/v) sodium azide. Cells were suspended in 5 ml cold water and transferred to six-well plate. Samples were incubated with 300 μl trioxsalen solution (200 μg/ml; in 100% ethanol) for 10 min and then irradiated by UVA (365 nm) for 10 min on ice; with total four repeats. Cells were spheroblasted in 5 ml yeast lysis buffer (1 M sorbitol, 100 mM EDTA, 14 mM β-mercaptoethanol and 5 mg Zymolyase®-100T) at 37 °C for 1 h. Spheroplasts were collected and then resuspended in 4 ml digesting buffer (800 mM guanidine HCl, 30 mM Tris-HCl [pH 8.0], 30 mM EDTA, 5%Tween-20 and 0.5% Triton X-100) with 100 μl RNase A (10 mg/ml) at 37 °C for 1 h. Two hundred microliter of proteinase K (20 mg/ml) were then added and incubated for 3 h at 50 °C following 30 °C overnight. The supernatants were diluted in an equal volume of equilibration buffer (750 mM NaCl, 50 mM MOPS [pH 7.0], 15% isopropanol and 0.15%Triton X-100) following load into the Genomic-tip 100/G (QIAGEN, Cat No.: 10243) anion exchange column (pre-equilibrated with 4 ml equilibration buffer). Columns were washed twice by 7.5 ml wash buffer (1.0 M NaCl, 50 mM MOPS [pH 7.0] and 15% isopropanol).Genomic DNA were eluted by 5 ml 50 °C elution buffer (1.25 M NaCl, 50 mM Tris-HCl [pH 8.5] and 15% isopropanol), following precipitated with 4 ml of isopropanol. Eight microgram of DNA samples were digested with restriction enzyme(s), were then precipitated and dissolved in 20 μl TE buffer following resolved by 1D gel (0.35% agarose, 1.5 V/cm for 19 h) and 2D gel (1% agarose, 6 V/cm for 5 h) electrophoresis in TBE buffer by Maxi Horizontal Gel Electrophoresis System (Major Science, ME20-25), followed by standard Southern blotting procedures. PCR-amplified DNA fragments of the indicated genomic regions were used to make probes for Southern blotting. The sequences of primers for PCR are listed in Supplementary Table 2. The probed membranes were exposed to Storage Phosphor Screens (GE Healthcare, BAS-IP MS 2040 E) for 4 days; signals were detected by Typhoon™ FLA 7000 Phosphorimager (GE Healthcare). Uncropped and unprocessed blots can be found in Source data.

**CAN1 mutation rate assay**. Eleven individual colonies were randomly selected and inoculated in YPD medium overnight at 30 °C. Each sample was diluted 100-fold into fresh medium with or without HU and cultures were allowed to reach stationary phase (≈30 h). Cells were then diluted and plated on either YPD to evaluate plating efficiency or S.C. minus arginine with 60 μg/ml of canavanine to select *can1* mutants. Growth were determined after incubation at 30 °C for 3 days. Mutation rates were measured by Lea-Coulson method of median, and significant differences were identified using a Mann–Whiney *U*-test.

**Reporting summary**. Further information on research design is available in the Nature Research Reporting Summary linked to this article.

## Data availability

The ChIP-seq and ATAC-seq data sets generated during and/or analyzed during the current study are available in the Gene Expression Omnibus with the accession code GSE133222. The uncropped and/or unprocessed blots and the source data underlying Figs. 1d, 5d, e and 7c–f; and supplementary Fig. 4a, c are provided as a Source Data file. A reporting summary for this article is available as Supplementary Information file. All other data supporting the findings of this study are available from the responding author upon reasonable request.

## Code availability

We used published algorithms. See Methods section for full description of each analysis including input data, library, and algorithm version used.

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

## Acknowledgements

We would like to thank David Allis for providing the yeast strains and plasmids and Rafal Donczew for the technical support of transcription frequency calculation. We wish to thank Wei-Chun Hsiao and You-Rou Liao for their technical supports on ChIP and ATAC-seq; and Marcus Calkins for editing and comments on this manuscript. We thank Meng-Chao Yao for his critical reading of this manuscript. S.Y.C. was supported by National Taiwan University and Academia Sinica Joint Program (NTU-AS-106R104509). S.B. is an investigator of the Howard Hughes Medical Institute. This work was supported by grants from National Institute of Health, USA (R01GM058465) to T.T., Ministry of Science and Technology, Taiwan (MOST105-2320-B-002-026-MY3) to Y.-C.L. (MOST 105-2320-B-001-023-MY3) to C.-F.K., and Academia Sinica (AS-TP-107-ML06 and AS-108-TP-L07) to C.-F.K.

## Author contributions

S.Y.C., Y.-C.L., and C.-F.K. conceived the project. S.Y.C. performed most of experiments. J.J.L. contributed reagents to and supervised R-loop experiments. C.-H.T. and H.-K.T. analyzed ATAC-seq data. C.-F.K. and S.C. performed the ChIP-seq experiments. S.B. and T.T. supervised genome-wide data analysis. S.Y.C., S.C., S.B., T.T., Y.-C.L., and C.-F.K. wrote the paper.

## Competing interests

The authors declare no competing interests.
