## [Peer Review File · Nature Communications]

Reviewers' comments:

Reviewer #1 (Remarks to the Author):

In this manuscript, the authors first show that reducing H3K4 methylation suppresses HU sensitivity of budding yeast *rad53* mutants and induction of (global) gH2A levels. Next, they show that upon HU exposure, loss of H3K4 methylation restores DNA pol 2 binding to chromatin and replication progression in *rad53* mutant cells. 2D gel analysis suggests that H3K4 methylation allows for fork reversal in these cells. Furthermore, introduction of a highly transcribed gene in two orientations next to an early autonomously replicating sequence to artificially induce head-on and co-directional transcription-replication conflict confirms that H3K4 methylation promotes fork impediments. The authors then show that in the absence of H3K4 methylation, replication appears to be faster, supportive of their model that H3K4 methylation may serve to decelerate replication in areas with high transcriptional activity. Finally, the authors show that elevated transcription leads to higher mutation rates, which is increased upon HU-induced replication stress and loss of RAD53 and H4K4 methylation.

Overall, I find this is an interesting and relevant paper that describes a potential function for transcription-associated H3K4 methylation in protecting genome integrity by regulating (speed of) replication fork progression. The manuscript is clearly written (few typos still exist), clearly understandable (also to the non-specialist) and appears to be experimentally sound and convincing. I have no major concerns and think that the manuscript is suitable for publication.

Minor concerns:

For their model, it would be more convincing if the authors could show that under conditions of low HU (<25 mM)/replication stress and normal transcription, when they observe suppression of the *rad53* mutant phenotype, that under these condition this 'rescued phenotype' comes at the expense of genome stability. In Figure 6, the authors show elevated mutation rates in the *CAN1* gene, but a real effect of loss of H3K4 methylation in a *rad53* background is only observed after exposure to high (100 mM) levels of HU and artificially boosting transcription of the gene. Furthermore, it is unclear why, if genomic instability is increased, there appears to be less H2A phosphorylation upon loss of H3K4 methylation in *rad53* mutant cells (Fig 1e). Please discuss this.

In several experiments, where the authors study the effect of loss of H3K4 methylation, it is assumed that H3K4 is indeed methylated at the studies genomic locations. For instance at *ARS305* in Fig 3e and in the experiments shown in Figure 4. Can the authors actually show (by e.g. ChIP) that turning on transcription of *LRE1* also leads to elevated H3K4 methylation, in line with the idea that this H3K4 methylation obstructs the replication fork progression? Also, for the *CAN1* gene, shown in Fig 6b, it is assumed that H3K4me levels increase after boosting transcription. The authors could experimentally address this assumption, or at least discuss this more clearly.

lines 208-211 Fig 4c should be 4d.

Reviewer #2 (Remarks to the Author):

The present work focuses on how histone methylation affects DNA replication fork progression during HU treatment. By suppressing the lethality of a *rad53* checkpoint kinase mutation during HU treatment with H3 mutations, the authors demonstrate that the extent of methylation of histones drives sensitivity to HU in *rad53* cells. Using genome-wide analyses of ChIP-Seq experiments, the authors find that replication fork stalling is increased at highly transcribed genes in *rad53* cells during HU treatment. Furthermore, they show quite nicely that the degree of stalling is dependent on the orientation of transcription with respect to replication.

The conclusion that HU sensitivity is independent of R-loops is misleading. Supplementary Figure 2 clearly shows that the double RNase H deletion cells are at least twice as sensitive to HU in the absence of RAD53 and this is rescued when H3 methylation is modulated. This result is consistent with the idea that replication-transcription conflicts are worse in cells lacking RAD53. Can the authors clarify what they mean by their interpretation that their observations are independent of R-loops?

The data are quite strong, the main concern however is the interpretation of the results as it pertains to the model. How can histone methylation serve as a “speed bump” to slow replication as it approaches a highly transcribed gene when these chromatin markers are within the gene body? Further, is the histone occupancy altered in H3K4 mutants? The authors should verify this using ATAC-Seq or similar methods.

Presumably, if the gene were highly transcribed, there would be just as much probability of the replication fork encountering an RNAP complex as a methylated histone. Under the provided model, the reason for increased fork stalling in rad53- cells during HU treatment would be due to a replisome complex interacting with an RNAP complex without being signaled via methylated histone markers. An alternative explanation is that during HU treatment, RAD53 signaling is needed for replication through methylated histones, independent of a direct interaction with RNAP. Can the authors distinguish between these two models? For example, if RNAP was modified such that it is less stable on DNA, would methylation of H3 still effect replisome progression during HU treatment?

The mutation rate data are intriguing. However, the interpretation that the increased rate of mutation is due to decreased fork fidelity during HU treatment is not supported by the data. CAN1 reversion can result from a variety of mutagenic events not limited by replicative fork errors. For example, it was previously demonstrated that increased mutation rates due to replication-transcription conflicts are mediated by the transcription-coupled DNA repair pathway and error-prone Y-family DNA polymerases (Work by originally Sue Jink-Robertson's and later the Merrikh group put forth these alternative models that should be discussed here).

Lastly, the authors do not reference the fundamental papers on replication-transcription conflicts. The foundation of the field was laid out by Andres Aguilera in yeast, and Benedicte Michel, Sarah French, as well as Houra Merrikh in bacteria. There is important work on conflicts from Sergei Mirkin that should also be included. Furthermore, the authors only reference a paper from the Cimprich lab when discussing R-loops. They should at the very least include Lang et al., which was published simultaneously.

Chong et al., " H3K4 methylation at active genes mitigates transcription-replication conflicts during replication stress".

Point-by-point Response to Reviewer Comments

*Please note that all changes in the manuscript text file are highlighted as bold and indicated by line # in the responses to the reviewer comments.

The major changes in the revised manuscript are the new Fig. 5. The previous Fig. 5 and Fig. 6 are now Fig. 6 and Fig. 7 respectively. To reduce the complexity, the model is moved into the new Fig. 8.

We thank the reviewers for their thoughtful and insightful comments on our manuscript. We have addressed their concerns and suggestions, as detailed below. The revised manuscript includes new experiments as well as changes to the text. Below are responses to each reviewer's comment:

Reviewer #1

Overall, I find this is an interesting and relevant paper that describes a potential function for transcription-associated H3K4 methylation in protecting genome integrity by regulating (speed of) replication fork progression. The manuscript is clearly written (few typos still exist), clearly understandable (also to the non-specialist) and appears to be experimentally sound and convincing. I have no major concerns and think that the manuscript is suitable for publication.

We are very grateful for reviewer #1's enthusiasm for our work.

Minor concerns:

#1. *For their model, it would be more convincing if the authors could show that under conditions of low HU (<25 mM)/replication stress and normal transcription, when they observe suppression of the rad53 mutant phenotype, that under these condition this 'rescued phenotype' comes at the expense of genome stability.*

To address the reviewer's question, we measured mutation rates in WT, H3K4A and *rad53*-H3K4A cells exposed to 5 mM HU. At this concentration of HU, *rad53* cells cannot survive, so the mutation rate in this background was not determined. The results showed that the mutation rate of *rad53*-H3 cells under 5 mM HU was similar with that of *RAD53*-H3 cells; however, the mutagenesis in *rad53*-H3K4A cells was dramatically elevated under the same conditions (**new Fig. 7e**). With this additional assay, we confirmed that loss of H3K4 methylation indeed rescues *rad53* viability under HU stress, with the additional consequence of increasing genome instability.

#2. In Figure 6, the authors show elevated mutation rates in the *CAN1* gene, but a real effect of loss of H3K4 methylation in a *rad53* background is only observed after exposure to high (100 mM) levels of HU and artificially boosting transcription of the gene.

Please note that the mutation rates of *rad53*-H3 and *rad53*-H3K4A were not tested in 100 mM HU because cells with *rad53* mutation cannot survive in high HU concentrations. Our interpretation of the effect of losing H3K4me in the *rad53* background is discussed in detail in response to the comment **#5** from reviewer **#2** below.

We have moved these data (previously in **Fig. 6b**) to **Fig. 7f** of our revised manuscript.

#3. Furthermore, it is unclear why, if genomic instability is increased, there appears to be less H2A phosphorylation upon loss of H3K4 methylation in *rad53* mutant cells (Fig 1e). Please discuss this.

In the text, we explain that γ H2A likely reflects the amount of fork stalling, because in the presence of less than 25 mM HU, forks in *rad53* cells does not collapse soon after origin firing. Therefore, the significant decrease of γ H2A in HU-treated *rad53*-mutants with deleted Set1C subunits should reflect a reduction of stalled forks. We have clarified this point in lines 104-109.

#4. In several experiments, where the authors study the effect of loss of H3K4 methylation, it is assumed that H3K4 is indeed methylated at the studies genomic locations. For instance at *ARS305* in Fig 3e and in the experiments shown in Figure 4. Can the authors actually show (by e.g. ChIP) that turning on transcription of *LRE1* also leads to elevated H3K4 methylation, in line with the idea that this H3K4 methylation obstructs the replication fork progression? Also, for the *CAN1* gene, shown in Fig

6b, it is assumed that H3K4me levels increase after boosting transcription. The authors could experimentally address this assumption, or at least discuss this more clearly.

We performed ChIP-qPCR to measure the H3K4me3 levels in the *pMET25-3HA-LRE1* gene under the conditions of activation or repression; levels were also measured in the *CAN1* gene with its native promoter or a strong *TEF1* promoter. The level of H3K4me3 was surveyed using three pairs of primers at the transcription start site (P1), the middle (P2) and the end (P3) of genes. The results showed that activation of *pMET25-3HA-LRE1* and boosting transcription of *CAN1* by *pTEF1* both lead to elevation of H3K4me3. The results are now shown in **Supplementary Fig. 4c** and **Fig. 7c**, respectively.

#5. lines 208-211 Fig 4c should be 4d.

We apologize for this mistake, which has been corrected. Additionally, this panel is now Fig. 4g in the revised manuscript.

Reviewer #2

The present work focuses on how histone methylation affects DNA replication fork progression during HU treatment. By suppressing the lethality of a rad53 checkpoint kinase mutation during HU treatment with H3 mutations, the authors demonstrate that the extent of methylation of histones drives sensitivity to HU in rad53 cells. Using genome-wide analyses of ChIP-Seq experiments, the authors find that replication fork stalling is increased at highly transcribed genes in rad53 cells during HU treatment. Furthermore, they show quite nicely that the degree of stalling is dependent on the orientation of transcription with respect to replication.

We thank this reviewer for his/her positive comment on our data set showing the dependence of replication fork stalling on transcription orientation.

#1. The conclusion that HU sensitivity is independent of R-loops is misleading. Supplementary Figure 2 clearly shows that the double RNase H deletion cells are at least twice as sensitive to HU in the absence of RAD53 and this is rescued when H3 methylation is modulated. This result is consistent with the idea that replication-transcription conflicts are worse in cells lacking RAD53. Can the authors clarify what they mean by their interpretation that their observations are independent of R-loops?

We thank the reviewer for taking note of this issue and agree that our data show that deletion of RNase H reduces the viability of *rad53* cells in HU; thus, our original statement regarding the interaction between RNase H and *RAD53* was inaccurate, and we have corrected the text accordingly. However, our results in Supplementary Fig. 2a clearly show that the loss of H3K4 methylation is able to partially rescue HU sensitivity of *rad53* cells **either in the presence or absence of RNase H**. These results suggest that accumulation of R-loops (caused by deleting genes encoding the RNase H components, *RNH1* and *RNH201*) indeed sensitizes the *rad53* mutant to HU stress; however, the loss of H3K4me is able to rescue HU-treated *rad53* cells even in the presence of more R-loops. We thus argue that the effect of H3K4 methylation on *rad53*-HU sensitivity is at least partially independent of RNase H activity and R loops. We have corrected our description of the results and clarified our conclusion and reasoning in the revised manuscript at lines 114-117.

#2. The data are quite strong, the main concern however is the interpretation of the results as it pertains to the model. How can histone methylation serve as a “speed bump” to slow replication as it approaches a highly transcribed gene when these chromatin markers are within the gene body?

We apologize that our description of the “speed bump” model was not sufficiently clear. We propose that H3K4 methylation mitigates TRCs by slowing down the replisome complex as it passes through highly expressed ORFs, where this histone mark is abundant. Therefore, we did not intend to claim that this histone modification in a coding region would slow a replication fork before it reaches (i.e., “as it approaches”) the gene. We speculate that slowing the replication fork, especially within the 5' portion of the transcribed region of the gene (where H3K4me is enriched), gives active transcription complexes more time to complete transcriptional activities, thereby reducing the potential for encounters between transcription and replication machinery and preventing TRCs. We clarify and emphasize this point in lines 281-289.

#3. Further, is the histone occupancy altered in H3K4 mutants? The authors should verify this using ATAC-Seq or similar methods.

As the reviewer suggested, we applied ATAC-seq under 25 mM HU-stress to perform a genome-wide survey of chromatin accessibility, which also reflects nucleosome positioning. The results are presented in Fig. 3d, 3e and Supplementary

3. The analysis of Pearson's correlation indicates that the ATAC-seq profiles of TSS regions and ORC-bound origins are very similar between WT H3 and H3K4A mutants (in both *RAD53* and *rad53* cells). In addition, the nucleosome footprint of ATAC-seq pattern at TSS or origins was not influenced by H3K4 methylation. The description of ATAC-seq analysis begins at line 181.

#4. Presumably, if the gene were highly transcribed, there would be just as much probability of the replication fork encountering an RNAP complex as a methylated histone. Under the provided model, the reason for increased fork stalling in *rad53*- cells during HU treatment would be due to a replisome complex interacting with an RNAP complex without being signaled via methylated histone markers. An alternative explanation is that during HU treatment, *RAD53* signaling is needed for replication through methylated histones, independent of a direct interaction with RNAP. Can the authors distinguish between these two models? For example, if RNAP was modified such that it is less stable on DNA, would methylation of H3 still effect replisome progression during HU treatment?

In summary of the comment, we believe the reviewer would like to understand whether RNA Pol II transcription itself or histone methylation directly causes fork stalling in this process. We thank the reviewer pointing out this important question. We have now addressed the reviewer's question by performing an experiment that allows us to separate the effect of RNAP complex and H3K4me on replication progression (Fig. 5). We took advantage of the inducible *pMET25-LRE1* system (with deletion of the H3K4me demethylase *Jhd2* to slow the erasure of H3K4me). The *pMET25-LRE1* gene was activated in *rad53* mutants with either H3 or H3K4A mutation during pre-culture and G1-arrested in methionine (-) medium (*LRE1-on*) to allow the deposition of H3K4me. The *LRE1* gene was suppressed before G1-release and then released into methionine (+) medium (*LRE1-off*) with 25 mM HU. Under this condition, 2D gel analysis revealed the impact of H3K4me on replisome progression without (or with a basal level of) interference of RNAP. Hence, in Fig.5, we added (1) a schematic of the procedure; (2) HU sensitivity of *jhd2Δ rad53* double mutants; (3) H3K4me pattern of *jhd2Δ*; (4) *pMET25-LRE1* expression levels after methionine was added to repress transcription; (5) H3K4me3 deposition status and (6) 2D gel analysis to show the fork status of *rad53* cells encountering pre-deposited H3K4me on *LRE1* gene in the absence (or with a basal level) of transcription activity. From the results, we conclude that H3K4 methylation plays a direct role in impeding replication progression in HU-treated *rad53* mutants. The description of Fig. 5 begins at line 257.

#5. The mutation rate data are intriguing. However, the interpretation that the increased rate of mutation is due to decreased fork fidelity during HU treatment is not supported by the data. CAN1 reversion can result from a variety of mutagenic events not limited by replicative fork errors. For example, it was previously demonstrated that increased mutation rates due to replication-transcription conflicts are mediated by the transcription-coupled DNA repair pathway and error-prone Y-family DNA polymerases (Work by originally Sue Jink-Robertson's and later the Merrikh group put forth these alternative models that should be discussed here).

We acknowledge that it is possible the increased mutation rate seen in this experiment may have resulted from defects in transcription-coupled repair. Regardless of the exact mechanism of action, the major point we are trying to make with this figure is not to define the exact cause of the increased mutation rate, but to argue that loss of H3K4me-dependent replication fork slow-down has a long-term cost of increased mutation rate. We have emphasized this point with a new experiment (Fig. 7e) suggested by Reviewer #1 (see reviewer #1, comment #1). In this revised manuscript, we have discussed the potential roles of H3K4me in promoting accurate DNA synthesis (lines 403-412). The possible mechanisms may involve interactions between the "replisome" complex and H3K4me decorated chromatin template and how these interactions elevate DNA synthesis accuracy by (1) fine-tuning the biochemical processes of replication machinery, (2) by promoting transcription-coupled repair, or (3) by reducing chromatin torsional stress during TRCs. We have referenced **Million-Weaver et al, PNAS (2015)**, **Kim et al, MCB (2010)** with regard to transcription-coupled repair.

#6. Lastly, the authors do not reference the fundamental papers on replication-transcription conflicts. The foundation of the field was laid out by Andres Aguilera in yeast, and Benedicte Michel, Sarah French, as well as Houra Merrikh in bacteria. There is important work on conflicts from Sergei Mirkin that should also be included. Furthermore, the authors only reference a paper from the Cimprich lab when discussing R-loops. They should at the very least include Lang et al., which was published simultaneously.

We thank the reviewer for pointing out that we missed important references and for suggesting several labs that are important to include. We have added 15 new references in the Introduction and Discussion to strengthen and support our views/data. These citations are grouped into four topics, shown below:

To summarize important findings that demonstrate transcription as one factor to alter genome stability, which is a fundamental finding in this field, we reference:

1. Gaillard H, Aguilera A. Transcription as a Threat to Genome Integrity. *Annual Review of Biochemistry*, Vol 85 **85**, 291-317 (2016).
2. Vilette D, Ehrlich SD, Michel B. Transcription-induced deletions in plasmid vectors: M13 DNA replication as a source of instability. *Mol Gen Genet* **252**, 398-403 (1996).
3. Kim N, Jinks-Robertson S. Transcription as a source of genome instability. *Nature Reviews Genetics* **13**, 204-214 (2012).
4. Mirkin EV, Mirkin SM. Replication fork stalling at natural impediments. *Microbiol Mol Biol R* **71**, 13-35 (2007).

We realize that most knowledge of TRCs was gained by studies in bacteria. Therefore, we included TRCs studies in bacteria by referencing the following important papers:

1. Merrikh H, Zhang Y, Grossman AD, Wang JD. Replication-transcription conflicts in bacteria. *Nat Rev Microbiol* **10**, 449-458 (2012).
2. Lang KS, *et al.* Replication-Transcription Conflicts Generate R-Loops that Orchestrate Bacterial Stress Survival and Pathogenesis. *Cell* **170**, 787-+ (2017).
3. Paul S, Million-Weaver S, Chattopadhyay S, Sokurenko E, Merrikh H. Accelerated gene evolution through replication-transcription conflicts. *Nature* **495**, 512-+ (2013).
4. Boubakri H, de Septenville AL, Viguera E, Michel B. The helicases DinG, Rep and UvrD cooperate to promote replication across transcription units in vivo. *EMBO J* **29**, 145-157 (2010).
5. De Septenville AL, Duigou S, Boubakri H, Michel B. Replication fork reversal after replication-transcription collision. *PLoS Genet* **8**, e1002622 (2012).

We adopted strategies and concepts (using *pTEF1* to promote gene expression) from the following papers, and found similar outcomes in our mutation rate assay:

1. Datta A, Jinksrobertson S. Association of Increased Spontaneous Mutation-Rates with High-Levels of Transcription in Yeast. *Science* **268**, 1616-1619 (1995).
2. Kim N, Abdulovic AL, Gealy R, Lippert MJ, Jinks-Robertson S. Transcription-associated mutagenesis in yeast is directly proportional to the level of gene expression and influenced by the direction of DNA replication. *DNA Repair* **6**, 1285-1296 (2007).

Additionally, we include papers that documented the idea that head-on TRCs are more detrimental to genome stability, which is consistent with our HO/CD- TRCs study in HU-treated *rad53* cells (Fig. 4g). Therefore, we referenced the following impressive studies to support our findings:

1. Lang KS, *et al.* Replication-Transcription Conflicts Generate R-Loops that Orchestrate Bacterial Stress Survival and Pathogenesis. *Cell* **170**, 787-+ (2017).
2. Paul S, Million-Weaver S, Chattopadhyay S, Sokurenko E, Merrikh H. Accelerated gene evolution through replication-transcription conflicts. *Nature* **495**, 512-+ (2013).
3. French S. Consequences of replication fork movement through transcription units in vivo. *Science* **258**, 1362-1365 (1992).
4. Liu B, Alberts BM. Head-on Collision between a DNA-Replication Apparatus and Rna-Polymerase Transcription Complex. *Science* **267**, 1131-1137 (1995).

REVIEWERS' COMMENTS:

Reviewer #1 (Remarks to the Author):

In this revised manuscript, the authors have performed several additional experiments that further strengthen its overall conclusions. They have satisfactorily addressed my concerns. In my opinion, this is an original and highly interesting manuscript suitable for publication.

Few typo's still exist:

The labeling of Fig 4e is incorrect, this should be 'Met(-)' according to the text and legend.

Line 75 depedent should be dependent

Line 97 introducing should be introduced

Reviewer #2 (Remarks to the Author):

The authors have addressed my concerns very nicely.

Point-by-point Response to Reviewer Comments

REVIEWERS' COMMENTS:

Reviewer #1 (Remarks to the Author):

In this revised manuscript, the authors have performed several additional experiments that further strengthen its overall conclusions. They have satisfactorily addressed my concerns. In my opinion, this is an original and highly interesting manuscript suitable for publication.

Few typo's still exist:

The labeling of Fig 4e is incorrect, this should be 'Met(-)' according to the text and legend.

Line 75 depedent should be dependent

Line 97 introducing should be introduced

We are very grateful for the Reviewer #1's enthusiasm for our work. We also want to thank the Reviewer#1 read our manuscript very carefully. These typos were fixed.

Reviewer #2 (Remarks to the Author):

The authors have addressed my concerns very nicely.

In sum, we thank the reviewers for their time in reviewing this manuscript. The comments, suggestions, recommendations greatly contributed to improve our manuscript significantly.